# DC4GS: Directional Consistency-Driven Adaptive Density Control for 3D Gaussian Splatting

**Moonsoo Jeong**[1]     **Dongbeen Kim**[2]     **Minseong Kim**[3]     **Sungkil Lee**[1,2,3,*]

[1]Department of Electrical and Computer Engineering, Sungkyunkwan University, South Korea
[2]Department of Computer Science and Engineering, Sungkyunkwan University, South Korea
[3]Department of Immersive Media Engineering, Sungkyunkwan University, South Korea
{moonsoo101, rlaehdqls021, leon0106, sungkil}@skku.edu

## Abstract

We present a Directional Consistency (DC)-driven Adaptive Density Control (ADC) for 3D Gaussian Splatting (DC4GS). Whereas the conventional ADC bases its primitive splitting on the magnitudes of positional gradients, we further incorporate the DC of the gradients into ADC, and realize it through the angular coherence of the gradients. Our DC better captures local structural complexities in ADC, avoiding redundant splitting. When splitting is required, we again utilize the DC to define optimal split positions so that sub-primitives best align with the local structures than the conventional random placement. As a consequence, our DC4GS greatly reduces the number of primitives (up to 30% in our experiments) than the existing ADC, and also enhances reconstruction fidelity greatly.

## 1 Introduction

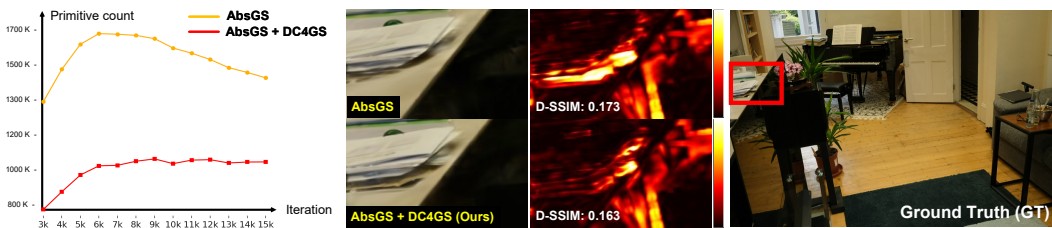

(a) Evolution of primitive counts during training                (b) Rendering quality improvement in high-frequency details

Figure 1: Comparison of AbsGS [38] and our Directional Consistency-driven density control (DC4GS) in terms of (a) primitive counts during training and (b) reconstruction quality. DC4GS saturates much earlier (here, 6k iterations) than the AbsGS does. Upon convergence, it achieves a significant reduction in primitive counts (here, 30%) and high-quality splits as well, resulting in much higher reconstruction quality (b) for high-frequency details (here, see the red inset).

*3D Gaussian Splatting* (3DGS) can synthesize high-quality renderings well, even from *sparse* point clouds [14]. Its efficiency stems from its adaptive density control (ADC) that leverages 2D positional gradients, derived from reconstruction loss, indicating how effectively the observations of the input signals are represented. When the primitives in fact are likely to require finer details (i.e., over-reconstruction), the ADC spawns new primitives by splitting coarse ones where they matter most.

---

*Corresponding author.
  Code is available at https://github.com/cgskku/dc4gs.

39th Conference on Neural Information Processing Systems (NeurIPS 2025).

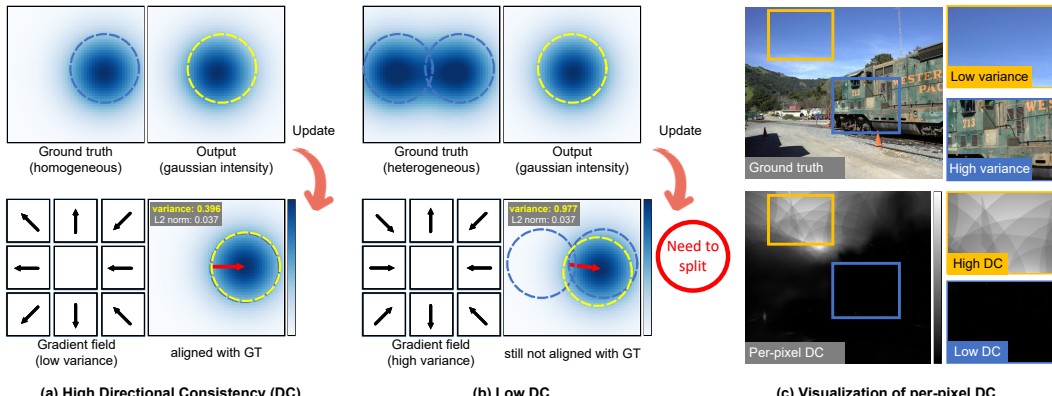

Figure 2: Visualization of how DC reflects structural complexities. In (a), a Gaussian can be aligned with a single-peak ground truth (GT) using a simple shift, which is indicated by a low angular variance (i.e., high DC). In contrast, (b) requires splitting due to the mismatch from the two-peak GT. This can be reflected in its divergent gradients (i.e., low DC), but not in the gradient magnitudes. (c) visualizes the per-pixel DC values in a real-world image, distinguishing directionally coherent (e.g., the sky) and incoherent (the train) regions well.

The positional gradients include valuable information of reconstruction, but the majority of the existing 3DGS-based methods still take only their magnitude (L2 norm) into account [14, 38, 40, 41, 44]. The amount of information available from the magnitude is limited in representing the distributions of individual primitives, and often leads to redundant splits where they are actually not required (e.g., low-frequency or already well-refined areas). Furthermore, split positions are randomly chosen and thereby disregard local structural cues, which often result in misaligned or overlapping sub-primitives.

In order for ADC to better reflect the homogeneity of the local structures, we exploit the directional distributions of the positional gradients as well as their magnitudes. When the directions of the gradients are coherent, the regions encompassing them are likely to be simpler, not requiring splitting. In contrast, regions with incoherent gradients require improving details. This approach can take advantages both of primitive counts and reconstruction quality (see Fig. 1 for example). To this end, we introduce *Directional Consistency* (DC) that captures the spatial coherence of the directional patterns of positional gradients within Gaussians. For its simple yet effective realization, our implementation uses the angular variance of gradient directions across pixels. Fig. 2 exemplifies the two Gaussian patterns whose gradient magnitudes are identical, but their DCs are different.

Building on this insight, we use DC as a versatile criterion in determining whether to split a primitive and where to place its sub-primitives. To guide the splitting, we incorporate DC into the existing ADC criteria, which better detects primitives in structurally complex regions. For split placement, we present a DC-guided split, splitting a primitive where the DCs of its sub-primitive candidates are maximized and the structural complexity each sub-primitive represents is minimized. By leveraging the DC cue in both decision and placement of splitting, we achieve higher reconstruction fidelity with much fewer primitives than the previous simpler approaches.

To summarize, the contributions of this paper can be listed in what follows:

- introduction of *Directional Consistency* (DC) as a criterion to better capture the structural complexity of local positional gradients in 3DGS;
- a DC-guided split decision so that structurally complex regions are better selected;
- a DC-guided split scheme so that the DCs of resulting sub-primitives are maximized and thereby their structures are best distinguished.

## 2 Related work

### 2.1 3D Gaussian splatting

The 3DGS [14] has emerged as a real-time alternative to Neural Radiance Fields (NeRFs) [24] that remains computationally intensive despite extensive advances in acceleration [5, 6, 9, 25, 28, 33, 39], anti-aliasing [1–3], and robustness [17, 34]. Unlike NeRFs, 3DGS enables efficient rendering by directly rasterizing explicit 3D Gaussian primitives without requiring dense regular grids/fields. However, 3DGS suffers from aliasing, blurring, and high-frequency detail loss in reconstruction [7].

To address aliasing from mismatched scale and insufficient filtering, several 3DGS-based methods [19, 21, 32, 40] introduce multiscale filtering and analytic rasterization to reduce artifacts and improve sharpness.

To better preserve high-frequency details and mitigate blurring caused by over-reconstruction, FreGS [42] applies progressive Fourier-space regularization to emphasize texture-rich structures, and BAGS [26] learns spatial blur kernels to recover over-smoothed details.

Geometric inconsistencies in 3DGS have also been addressed. Optimal-GS [13] reduces projection errors through local affine approximations, and Scaffold-GS [22] adopts anchor-based hierarchical representations, where neural anchors spawn offset primitives to model local structures, preserving global structural coherence.

### 2.2 Adaptive density control for 3D Gaussian splatting

The conventional ADC scheme in the 3DGS refines Gaussian primitives by iteratively splitting or cloning them based on positional gradient magnitudes. However, the ADC is sensitive to hyperparameters and often causes redundant splits and spatial misalignment.

To improve the selectivity of the ADC, several methods refine the processing of view-space positional gradients. AbsGS [38] and GOF [41] mitigate gradient cancellation by using the magnitude of homodirectional (absolute) gradients and the norm of positional gradients. Pixel-GS [44] introduces pixel-aware gradients by weighting contributions based on pixel coverage and view-space depth, densifying under-sampled regions.

The ADC was reformulated from alternative optimization perspectives. Revising Densification [29] guides growth via pixel-level error propagation, while a Markov Chain Monte Carlo-based method [15] interprets the ADC as stochastic sampling using Stochastic Gradient Langevin Dynamics [4] to reallocate inactive Gaussians without explicit densification.

Memory-efficient ADC is explored in Compact-3DGS [18], which prunes primitives using volume masks, and GES [10], which applies frequency-based pruning via generalized exponents. Recently, Localized Points Management (LPM) [37] improves ADC by identifying error-contributing zones under multi-view constraints and applying localized densification with opacity reset.

While the existing 3DGS-based methods often over-refine homogeneous regions and misalign sub-primitives, our work incorporates the DC to evaluate the homogeneity of Gaussians, enabling more selective splitting and better alignment with scene structure. This results in significant improvements in both the quality and storage efficiency.

## 3 Preliminary: adaptive density control in 3D Gaussian splatting

In this section, we briefly review the ADC in the standard 3DGS [14]. The 3DGS is typically initialized with sparse points derived from Structure-from-Motion (SfM) [30, 31], and densifies Gaussians via the ADC. This process entails evaluating view-space positional gradients $\partial L/\partial \mu'$, where $L$ is reconstruction loss and $\mu'$ is the center of a projected 2D Gaussian. The gradients are used to identify regions requiring higher details. The ADC criterion $\nabla_{\mu'} L$ is computed every hundred optimization iterations, and is defined as the average magnitude of the positional gradients:

$$\nabla_{\mu'_i} L = \frac{1}{\nu} \sum_{v=1}^{\nu} \|\frac{\partial L_v}{\partial \mu'_{i,v}}\|, \tag{1}$$

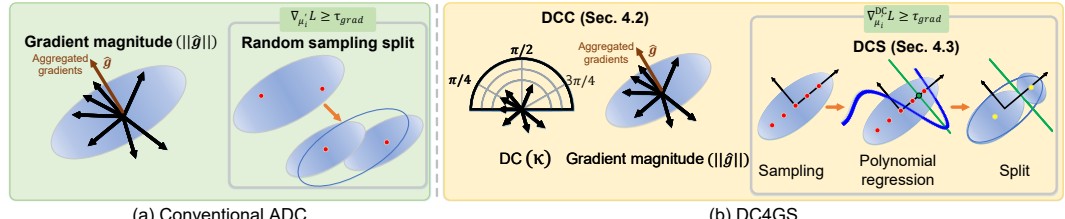

(a) Conventional ADC  (b) DC4GS

Figure 3: Comparison of the conventional ADC scheme (e.g., 3DGS [14], AbsGS [38], and Pixel-GS [44]) and our DC4GS. (a) Gaussian primitives are selected using the positional gradient magnitude-based criterion, and the new Gaussians are randomly spawned within the pre-split Gaussian. (b) Our DC-based split Criterion (DCC) further integrates the DC (the circular mean of the positional gradients) into the previous ADC criterion. Also, our DC-guided Split (DCS) better places the new sub-primitives so that their DCs are best distinguished.

where $\nu$ denotes the number of views in which a Gaussian $i$ is observed, and $L_v$ is the loss for the view $v$. When $\nabla_{\mu_i'} L$ of a Gaussian exceeds a threshold $\tau_p$ and its maximum scale $\|S_i\|$ surpasses $\tau_S$, the Gaussian $i$ is split, and produces $N$ sub-primitives ($N$=2 by default) by randomly sampling new centers within its coverage. If the gradient exceeds $\tau_p$, but $\|S_i\|$ is below $\tau_S$, the Gaussian is cloned in place. Gaussians with opacity below a given threshold are pruned out for efficiency.

We can extend not only the ADC criterion of the 3DGS in Eq. (1), but also the recent criteria still based on the gradient magnitude [38, 44], by incorporating DC into them. These extended criteria guide both the split decisions and sub-primitive placements. The details of how to define and apply our DC into the ADC are presented in Sec. 4.

## 4  DC4GS: Directional consistency-driven adaptive density control for 3DGS

The key idea of DC4GS is to quantify the directional coherence of positional gradients within a Gaussian primitive, which thereby enhances the gradient magnitude-based ADC schemes. The DC serves as an effective cue for assessing how homogeneous the region represented by a primitive is. We incorporate the DC into ADC to facilitate two key operations: 1) deciding whether a primitive should be split; 2) determining where to place sub-primitives when splitting is required.

To this end, we extend the existing densification criterion (Eq. (1)) by weighting it with the DC, which we refer to as the *DC-weighted split Criterion (DCC)*. When a DCC value is high, a single Gaussian primitive is likely to cover a heterogeneous region, which cannot adequately capture the region; this suggests refinement through splitting. Once a primitive is determined to be split, we also improve the placement of the newly spawned primitives; the conventional scheme places them randomly. We evaluate multiple candidate split locations within it and select the one that minimizes a DC-based cost. This divides a heterogeneous region into distinct sub-regions that are internally homogeneous. We referred this scheme as the *DC-guided Split (DCS)*.

Fig. 3 illustrates the differences between the conventional ADC and our DC4GS. Our DC4GS is readily compatible with the existing 3DGS-based pipelines, and thereby, can be easily plugged into their pipelines.

### 4.1  Directional consistency

The DC is computed from the positional gradients within a Gaussian, derived from reconstruction loss. Since the positional gradient magnitude also reflects how sensitive the loss is to positional changes, combining them together allows the 3DGS to better understand whether each primitive represents the regions well or not.

To quantify the DC within a Gaussian, we first assess the directional coherence of its positional gradients using a *circular mean* (or negative *circular variance*) [23] that is a statistical measure of angular dispersion of a set of unit vectors. Given a set of $N$ unit vectors $u_j$, its circular mean is defined as $C = \frac{1}{N} \sum_{j=1}^{N} u_j$, and its L2 norm $\|C\|$ reflects directional alignment. $\|C\| \approx 1$ implies

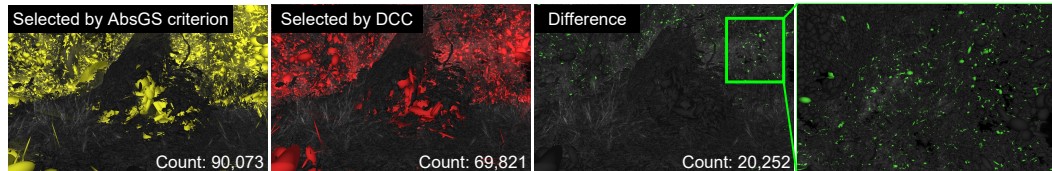

Figure 4: Comparison of split candidates selected by the criterion of AbsGS and DCC from the identical training states. After 14,900 training steps on the Stump scene using the 3DGS, Gaussians selected for splitting are visualized in yellow (AbsGS) and red (DCC). The difference (green) shows (potentially redundant) 20,252 Gaussians are not split by the DCC. Most of these differences are already of tiny sizes, suggesting limited structural gain from further splitting.

the vectors are aligned, and $\|C\| \approx 0$ implies angularly dispersed vectors. On the other hand, the circular variance is given by $V = 1 - \|C\| \in [0, 1]$, and represents directional inconsistency.

In our case, given a Gaussian $i$, we define its *directional consistency* $\kappa_i = \|C_i\|$. In the context of 3DGS, we evaluate the circular mean with respect to its positional gradients. For each pixel $j$ influenced by the Gaussian $i$, let $g_{i,j} = \partial L_j / \partial \mu'_i$ be the positional gradient and $u_{i,j} = g_{i,j}/\|g_{i,j}\|$ be its unit vector. The circular mean $C_i$ of the positional gradients is computed as:

$$C_i = \frac{1}{N} \sum_{j=1}^{N} u_{i,j} \in \mathbb{R}^2. \tag{2}$$

### 4.2 Directional consistency-weighted split criterion

Our DC can guide whether a Gaussian should be split. To this end, we introduce DC-guided split Criterion (DCC), which integrates the DC with the positional gradient magnitude. It identifies primitives that have a higher impact on the reconstruction and cover regions that cannot be well modeled by a single Gaussian.

To utilize DCC, we first aggregate the positional gradient magnitudes at pixels affected by Gaussian $i$ as $\hat{g}_i = \sum_j |g_{i,j}|$, following the homodirectional positional gradients accumulation in AbsGS [38]. We use this formulation by default, but $\hat{g}_i$ can be replaced with other alternatives, such as that used in 3DGS (Eq. (1)) or Pixel-GS [44]; we demonstrate their integrations in Sec. 6.

Then, the DCC for Gaussian $i$ can be defined as:

$$\nabla^{\text{DC}}_{\mu'_i} L = \frac{1}{\nu} \sum_{v=1}^{\nu} (1 - \kappa_{i,v}) \cdot \|\hat{g}_{i,v}\|, \tag{3}$$

where $\nu$ is the number of views in which the Gaussian is visible, and $\kappa_{i,v}$ denotes the DC evaluated for view $v$. We follow the per-view averaging scheme of the 3DGS [14], but augment the positional gradient magnitude with the complement of the DC, $1 - \kappa_{i,v}$, as a weighting factor. The Gaussian $i$ is selected for splitting, if its DCC, $\nabla^{\text{DC}}_{\mu'_i} L$, exceeds the gradient threshold $\tau_p$, and its maximum scale $\|S_i\|$ is smaller than the scale threshold $\tau_S$. Gaussians with high DC values, typically corresponding to homogeneous regions, are thus excluded from splitting.

As shown in Fig. 4, DCC well filters out refined primitives that are unlikely to benefit from a succeeding subdivision, and thereby, reduces the effective number of primitives selected for splitting compared to the gradient magnitude-based criterion.

### 4.3 Directional consistency-guided split

Our DCS improves the previous random sub-primitive placement by selecting split locations so that each sub-primitive is assigned to a distinct, locally homogeneous region. Since a single-peaked Gaussian is inherently suited for homogeneous regions, we guide the placement by evaluating the DC-based cost over candidate split locations, leading to sub-primitives being placed in regions with high DC.

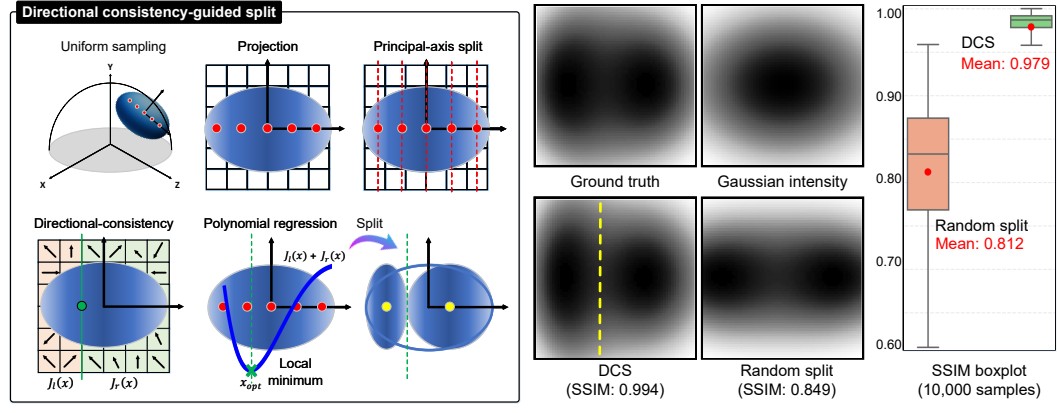

(a) Overview of DCS          (b) Comparison of DCS and random split in 2D

Figure 5: Overview and examples of the DCS. (a) Candidate points are uniformly sampled along the principal axis and projected onto the image plane. The split cost $J(x) = J_l(x) + J_r(x)$ is computed using the directional consistency and gradient magnitudes, and the polynomial regression is used to find the optimal split position $x_{\text{opt}}$ from the limited set of discrete samples. (b) In a 2D experimental example, our DCS well aligns the split along the regions structural change occurs, resulting in faithful reconstruction. The plot demonstrates quality statistics in over 10,000 randomized samples.

We constrain candidate split locations to lie along the principal axis of the Gaussian, defined as an axis having the largest scale in the anisotropic matrix $S$. This axis, $p_{\text{local}}$, is transformed to world coordinates as $p = R \cdot p_{\text{local}}$, using the rotation matrix $R$ of the Gaussian. The split is then performed orthogonally to this axis. Since the principal axis often spans multiple structures, a single elongated primitive exhibits large spatial variance, which increases blur and causes fine details to be smoothed out in structurally distinct regions. This axis-aligned sub-primitive placement mitigates such issues by reducing spatial spread and avoiding the overlaps commonly introduced by random placement, which could otherwise lead to structural discontinuities and incomplete coverage of the pre-split primitives.

To determine the optimal split location, we evaluate the DC-based cost over $N$ candidate locations, placed symmetrically along $p$ relative to the Gaussian center. At each candidate position $x$, a split line orthogonal to the projected 2D principal axis divides the primitive into left and right sub-regions. For each sub-region, we compute the DC, $\kappa(x)$, and the homodirectional gradient magnitude $\|\hat{g}(x)\|$. The sub-region costs are defined as $J_l(x) = (1 - \kappa_l(x)) \cdot \|\hat{g}_l(x)\|$ and $J_r(x) = (1 - \kappa_r(x)) \cdot \|\hat{g}_r(x)\|$, respectively. The total split cost $J(x)$ is: $J(x) = J_l(x) + J_r(x)$.

Minimizing the DC-based cost facilitates splitting at a point that divides a heterogeneous region into more homogeneous sub-regions, preventing reconstruction sensitivity from being concentrated in a single sub-primitive. Continuously evaluating $J(x)$ is costly in terms of computation and storage, and hence, we employ discrete sampling for efficiency. The overall procedure is illustrated in Fig. 5(a). In our implementation, we empirically set the samples of $N=5$. Finding the best candidates among the discrete samples can be straightforward, but may require too many samples for a complex junction. To improve this without dense sampling, we polynomially regress the samples [11]. This allows us to efficiently estimate the optimal split location $x_{\text{opt}}$ with a minimal number of per-primitive samples; see Appendix A.2 for details.

Notably, the DC-based cost aligns with our DCC in Eq. (3), reinforcing consistent reasoning in the decision to split and the placement of sub-primitives. Although the cost is evaluated before the actual split, minimizing it implicitly guides the sub-primitives toward homogeneous regions, which are less likely to require further splitting.

To validate the effectiveness of DCS, we compare it against random placement using 2D toy examples (see Fig. 5(b)). Here, DCS produces splits that more closely align with the ground truth, and demonstrates more consistent and accurate results over 10,000 randomly generated examples. Its practical effectiveness is demonstrated in Sec. 6.

**Algorithm 1** Optimization and Densification

$\|S\|$: maximum Gaussian scale, $\tau_p, \tau_S$: thresholds for $\nabla_{\mu'}^{DC} L$ and $\|S\|$, $N$: number of split candidates

---

$M, S, R, C, A \leftarrow$ InitAttributes()           $\triangleright$ Positions, Scales, Rotations, Colors, Opacities

$\nabla_{\mu'}^{DC} L \leftarrow 0$, $k \leftarrow 0$, $\nu \leftarrow 0$    $\triangleright$ DC-weighted split criterion, iteration counter, visibility counter

**while** not converged **do**

    $V, \hat{I} \leftarrow$ SampleTrainingView()          $\triangleright$ view-projection camera matrix, ground truth

    $I \leftarrow$ Rasterize($M, S, R, C, A, V$)

    $L \leftarrow$ Loss($I, \hat{I}$), $\frac{\partial L}{\partial \mu'} \leftarrow \nabla L$          $\triangleright$ backpropagation

    **for all** Gaussians $G_i$ visible in $V$ **do**          $\triangleright$ $i$: index of a Gaussian

        $\nu_i \leftarrow \nu_i + 1$, $\mu_i \leftarrow M_i$

        $\kappa_i \leftarrow$ EvalDirectionalConsistency($\frac{\partial L}{\partial \mu'_i}$)          $\triangleright$ Algo. 2

        $\hat{g}_i \leftarrow \sum_j |\frac{\partial L_j}{\partial \mu'_i}|$, $\nabla_{\mu'_i}^{DC} L \leftarrow \nabla_{\mu'_i}^{DC} L + (1 - \kappa_i) \cdot \|\hat{g}_i\|$  $\triangleright$ $j$: pixel position influenced by $G_i$

        $J_i \leftarrow J_i +$ EvalSplitCosts($\mu_i, S_i, R_i, V, N, \frac{\partial L}{\partial \mu'_i}$)          $\triangleright$ Algo. 3

    **if** IsRefinementIteration($k$) **then**

        **for all** Gaussians $G_i$ **do**

            $\nabla_{\mu'_i}^{DC} L \leftarrow (\nabla_{\mu'_i}^{DC} L)/\nu_i$

            **if** $\nabla_{\mu'_i}^{DC} L > \tau_p$ **and** $\|S_i\| > \tau_S$ **then**

                $x_{\text{opt}} \leftarrow$ PolynomialRegression($J_i, N$)

                SplitGaussian($x_{\text{opt}}, \mu_i, S_i, R_i, C_i, A_i$)          $\triangleright$ Algo. 5

            $\nabla_{\mu'_i}^{DC} L \leftarrow 0$, $J_i \leftarrow 0$, $\nu_i \leftarrow 0$

    $M, S, R, C, A \leftarrow$ Adam($\nabla L$), $k \leftarrow k + 1$          $\triangleright$ update parameters and increment iteration

---

# 5 Implementation details

Our DC4GS can be integrated seamlessly into the existing 3DGS pipelines. We implemented ours on top of the 3DGS [14], as well as its recent extensions: Pixel-GS [44] and AbsGS [38]. Algorithm 1 summarizes the procedure, where yellow-highlighted lines indicate the modifications to the previous ADC schemes. Clone and prune operations are identical to the baselines and thus omitted here.

For each visible Gaussian $i$, we evaluate its DC $\kappa_i$, accumulate its DCC $\nabla_{\mu'_i}^{DC} L$, and estimate its DC-based split cost $J_i$ over $N$ candidate split location samples along the principal axis. The primitive is selected for splitting if its DCC exceeds the gradient threshold $\tau_p$ and its maximum scale $\|S_i\|$ is smaller than the scale threshold $\tau_S$. Once selected for splitting, the optimal split location $x_{\text{opt}}$ is determined via the polynomial regression over $J_i$. Additional details are provided in Appendix A.1.

# 6 Experiments

## 6.1 Experimental details

**Datasets and metrics.** We evaluate our DC4GS on the three standard datasets: Mip-NeRF 360 [2] (five outdoor and four indoor scenes), and two scenes each from Tanks and Temples [16] and Deep Blending [12], following the 3DGS [14]. Every 8th frame is used for test. Rendering quality is evaluated by PSNR, SSIM [35], and LPIPS [43]. We also report memory requirements for 3DGS parameters and the number of primitives to assess storage efficiency.

**Experimental setup.** To ensure a fair comparison with the baselines [14, 38, 44], we adopt their training schedules, loss functions, and all hyperparameters, including the gradient threshold $\tau_p$ and the scale threshold $\tau_S$. As with the baselines, we halt the densification after 15K iterations. All the experiments are conducted on a single NVIDIA A6000 GPU with 48GB of memory.

## 6.2 Comparison with 3DGS-based methods

We compare our DC4GS with the aforementioned baselines by integrating it into their ADCs. Our comparisons with the baselines also include non-3DGS novel-view synthesis methods, including

Table 1: Quantitative comparison of the 3DGS, Pixel-GS, and AbsGS on real-world datasets without and with integrating DC4GS. The results denoted by '*' were excerpted from the 3DGS paper. The rest were obtained through our in-house experiments. The top score , second-highest score , and third-highest score are color-coded in red, orange, and yellow, respectively.

| Method | Mip-NeRF360 | | | | | Tanks&Temples | | | | | Deep Blending | | | | |
|---|---|---|---|---|---|---|---|---|---|---|---|---|---|---|---|
| | PSNR↑ | SSIM↑ | LPIPS↓ | Prim. | Mem. | PSNR↑ | SSIM↑ | LPIPS↓ | Prim. | Mem. | PSNR↑ | SSIM↑ | LPIPS↓ | Prim. | Mem. |
| Plenoxels * [8] | 23.080 | 0.626 | 0.463 | - | 2.1GB | 21.080 | 0.719 | 0.379 | - | 2.3GB | 23.060 | 0.795 | 0.510 | - | 2.7GB |
| iNGP-big * [25] | 25.590 | 0.699 | 0.331 | - | 48MB | 21.920 | 0.745 | 0.305 | - | 48MB | 24.960 | 0.817 | 0.390 | - | 48MB |
| Mip-NeRF360 * [2] | 27.690 | 0.792 | 0.237 | - | 8.6MB | 22.220 | 0.759 | 0.257 | - | 8.6MB | 29.400 | 0.901 | 0.245 | - | 8.6MB |
| 3DGS [14] | 27.414 | 0.812 | 0.218 | 3350K | 792MB | 23.655 | 0.844 | 0.179 | 1893K | 447MB | 29.394 | 0.898 | 0.248 | 2833K | 670MB |
| Scaffold-GS [22] | 27.651 | 0.810 | 0.226 | 5460K | 164MB | 24.018 | 0.851 | 0.174 | 2470K | 74MB | 30.252 | 0.904 | 0.255 | 1710K | 51MB |
| GES [10] | 27.040 | 0.796 | 0.248 | 1543K | 365MB | 23.640 | 0.842 | 0.191 | 930K | 220MB | 29.562 | 0.903 | 0.249 | 1606K | 380MB |
| LPM [37] | 27.589 | 0.820 | 0.212 | 3426K | 810MB | 23.878 | 0.847 | 0.183 | 1824K | 431MB | 29.483 | 0.901 | 0.245 | 2525K | 597MB |
| Pixel-GS [44] | 27.537 | 0.822 | 0.190 | 5622K | 1329MB | 23.759 | 0.853 | 0.151 | 4598K | 1087MB | 28.812 | 0.891 | 0.252 | 4623K | 1093MB |
| AbsGS [38] | 27.504 | 0.818 | 0.191 | 3149K | 744MB | 23.636 | 0.852 | 0.162 | 1332K | 315MB | 29.500 | 0.900 | 0.237 | 1961K | 463MB |
| AbsGS (60K iter.) | 27.539 | 0.817 | 0.189 | 3162K | 748MB | 24.039 | 0.855 | 0.156 | 1313K | 311MB | 29.275 | 0.895 | 0.240 | 1962K | 464MB |
| 3DGS+DC4GS | 27.486 | 0.814 | 0.217 | 2968K | 701MB | 23.786 | 0.845 | 0.179 | 1703K | 402MB | 29.565 | 0.901 | 0.245 | 2644K | 625MB |
| Scaffold-GS+DC4GS | 27.653 | 0.810 | 0.225 | 4790K | 144MB | 24.016 | 0.849 | 0.177 | 2080K | 62MB | 30.063 | 0.903 | 0.257 | 1350K | 40MB |
| GES+DC4GS | 27.090 | 0.797 | 0.248 | 1323K | 313MB | 23.515 | 0.841 | 0.194 | 816K | 193MB | 29.632 | 0.904 | 0.248 | 1487K | 352MB |
| LPM+DC4GS | 27.556 | 0.819 | 0.218 | 2712K | 641MB | 23.892 | 0.846 | 0.186 | 1502K | 355MB | 29.584 | 0.903 | 0.244 | 2350K | 555MB |
| Pixel-GS+DC4GS | 27.620 | 0.824 | 0.191 | 5009K | 1184MB | 23.930 | 0.855 | 0.150 | 4106K | 971MB | 29.181 | 0.896 | 0.246 | 4284K | 1013MB |
| AbsGS+DC4GS | 27.625 | 0.826 | 0.188 | 2615K | 618MB | 24.121 | 0.859 | 0.159 | 1093K | 258MB | 29.654 | 0.905 | 0.235 | 1499K | 354MB |

Mip-NeRF360 [2], iNGP-big [25], and Plenoxels [8]. All the baselines are either re-run using their official implementations with default settings, or taken directly from published results for fairness.

### 6.2.1 Quantitative comparison

Table 1 shows a quantitative analysis for the three datasets. Overall, when DC4GS is integrated into the baselines [14, 38, 44, 22, 10, 37], it consistently improves them to competitive or superior quality. For Scaffold-GS [22], which does not involve any explicit primitive splitting logic, we only apply our DC-weighted Split Criterion (DCC) and omit the DC-guided Split (DCS). In addition to the quality improvement, DC4GS significantly reduces the number of primitives and memory usage for 3DGS parameters; the largest reduction (on average 20%) is observed in combining our density control with AbsGS, and consistent savings for the other baselines are observed as well. Notably, the combination of DC4GS with AbsGS also yields the greatest improvement in reconstruction quality. DC4GS also achieves up to 11.5% fewer primitives with 3DGS, 11% with Pixel-GS, and 12–18% with Scaffold-GS, GES, and LPM, improving quality metrics. In contrast, simply extending AbsGS training to 60K iterations yields only marginal or even negative gains, indicating that the improvements of DC4GS stem from structural complexity-aware splitting rather than brute-force optimization or longer training (see Table 1).

### 6.2.2 Training and rendering efficiency comparison

Table 2 compares the training and per-frame rendering times between the baselines and their DC4GS-integrated models. Here, the rendering time is measured by averaging the time taken to render all test-set images for each dataset. The integration of the DC4GS introduces moderate training overhead due to the additional computation of the DC and DC-based cost evaluation. However, this additional cost is confined to training only. At inference time, DC4GS improves rendering efficiency by reducing the number of Gaussian primitives. The consistent speedups in rendering align with the reduced primitive counts, demonstrating the practical scalability of our DC4GS.

### 6.2.3 Qualitative comparison

Fig. 6 visually compares the improvements achieved by DC4GS. Compared to 3DGS and AbsGS, our DC4GS more faithfully preserves fine structures, such as the window frames, railings, and rods in the figure, where both the baselines often exhibit objectionable discontinuities. It also better maintains object boundaries (the third and last rows), which tend to be oversmoothed in the baselines. While AbsGS generally achieves high visual quality, it is likely to over-split already fine textures, such as the leaves in front of the windows or the grass near rocks (the fourth row), which can result in visual occlusions and structural artifacts. These observations indicate that DC4GS better preserves connected structures such as window frames and railings, which often appear fragmented or discontinuous in the baseline results. It also retains fine details in high-frequency textures, like the leaves or grass, without being occluded by background objects, which is frequently observed in the baselines.

Table 2: Comparison of training time and per-frame rendering time (ms) for the baselines and DC4GS-integrated models on the Mip-NeRF360, Tanks&Temples, and Deep Blending datasets.

| Method | Mip-NeRF360 | | Tanks&Temples | | Deep Blending | |
|---|---|---|---|---|---|---|
| | Training | Rendering (ms) | Training | Rendering (ms) | Training | Rendering (ms) |
| **3DGS** | **34m** | 10.472 | **19m** | 7.87 | **30m** | 9.646 |
| **3DGS+DC4GS** | 49m | **9.836** | 27m | **7.518** | 47m | **9.023** |
| **Scaffold-GS** | **1h 1m** | 9.705 | **29m** | 7.015 | **42m** | 6.298 |
| **Scaffold-GS+DC4GS** | **1h 1m** | **9.563** | 31m | **6.933** | 44m | **5.748** |
| **GES** | **32m** | 6.981 | **18m** | 5.593 | **40m** | 6.835 |
| **GES+DC4GS** | 42m | **6.698** | 22m | **5.441** | 45m | **6.571** |
| **LPM** | **38m** | 10.702 | **18m** | 7.143 | **30m** | 9.090 |
| **LPM+DC4GS** | 54m | **9.462** | 27m | **6.370** | 49m | **8.512** |
| **Pixel-GS** | **49m** | 17.163 | **35m** | 15.212 | **41m** | 14.271 |
| **Pixel-GS+DC4GS** | 1h 6m | **16.211** | 45m | **14.450** | 1h 1m | **13.469** |
| **AbsGS** | **37m** | 9.975 | **17m** | 6.175 | **27m** | 7.353 |
| **AbsGS (60K iter.)** | 1h 19m | 10.247 | 38m | 6.690 | 57m | 7.549 |
| **AbsGS+DC4GS** | 51m | **9.191** | 23m | **5.778** | 40m | **6.378** |

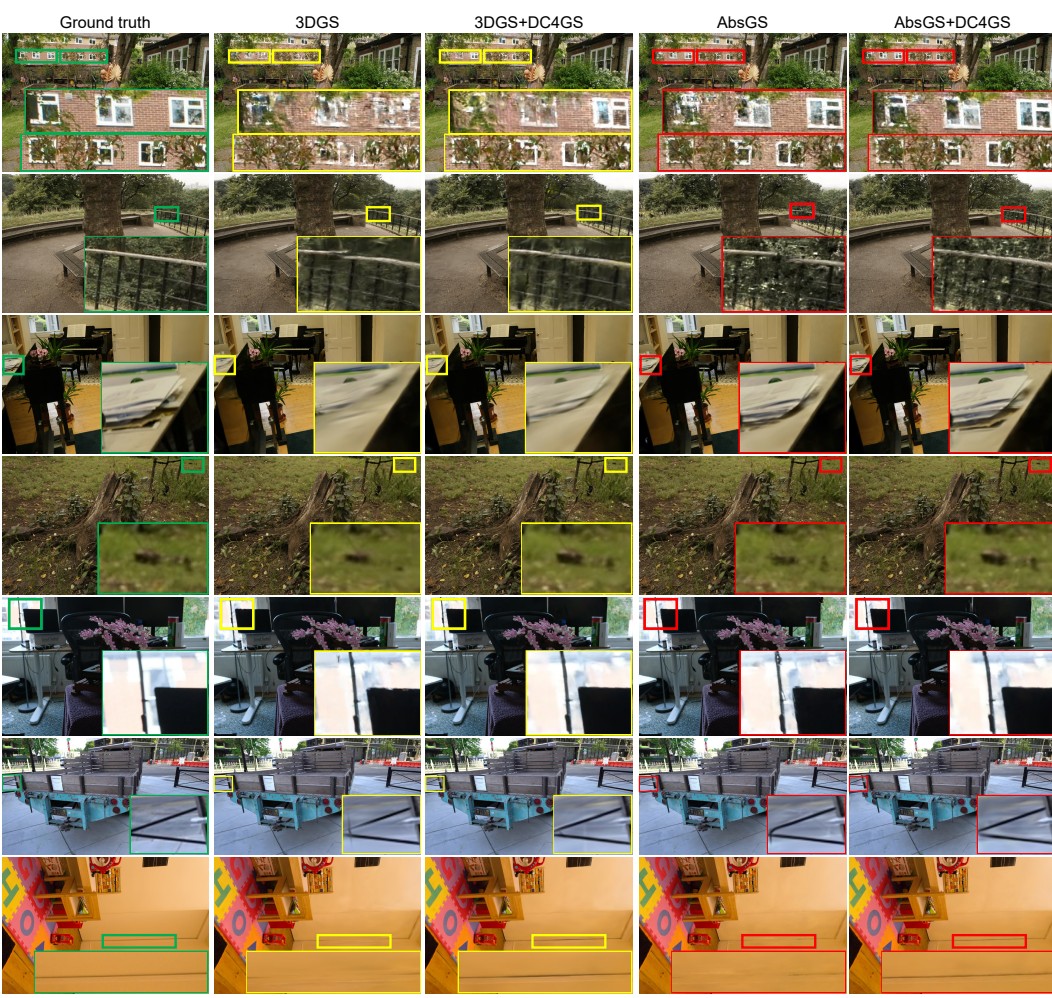

Figure 6: Qualitative comparison of the baseline methods (3DGS and AbsGS) without and with the integration of our DC4GS. The scenes, from top to bottom, are: Garden, Treehill, Room, Stump, and Bonsai from the Mip-NeRF360 dataset, Truck from the Tanks&Temples, and Playroom from Deep blending dataset.

Table 3: Ablation study evaluating DCC (Sec. 4.2) and DCS (Sec. 4.3) on the Mip-NeRF360, Tanks&Temples, and Deep Blending dataset. The AbsGS [38] is selected as the baseline.

| Baseline | DCC | DCS | Mip-NeRF360 | | | | | Tanks&Temples | | | | | Deep Blending | | | | |
|---|---|---|---|---|---|---|---|---|---|---|---|---|---|---|---|---|---|
| | | | PSNR ↑ | SSIM ↑ | LPIPS ↓ | Prim. ↓ | Mem. | PSNR ↑ | SSIM ↑ | LPIPS ↓ | Prim. ↓ | Mem. | PSNR ↑ | SSIM ↑ | LPIPS ↓ | Prim. ↓ | Mem. |
| ✓ | | | 27.504 | 0.818 | 0.191 | 3149K | 744MB | 23.636 | 0.852 | 0.162 | 1332K | 315MB | 29.500 | 0.900 | 0.237 | 1961K | 463MB |
| ✓ | ✓ | | 27.507 | 0.819 | 0.191 | 2861K | 676MB | 23.661 | 0.852 | 0.164 | 1197K | 283MB | 29.567 | 0.901 | 0.237 | 1799K | 425MB |
| ✓ | | ✓ | 27.587 | **0.826** | **0.187** | 2877K | 680MB | 24.018 | 0.857 | **0.158** | 1198K | 283MB | 29.591 | **0.905** | **0.235** | 1609K | 380MB |
| ✓ | ✓ | ✓ | **27.625** | **0.826** | 0.188 | **2615K** | 618MB | **24.121** | **0.859** | 0.159 | **1093K** | 258MB | **29.654** | **0.905** | **0.235** | **1499K** | 354MB |

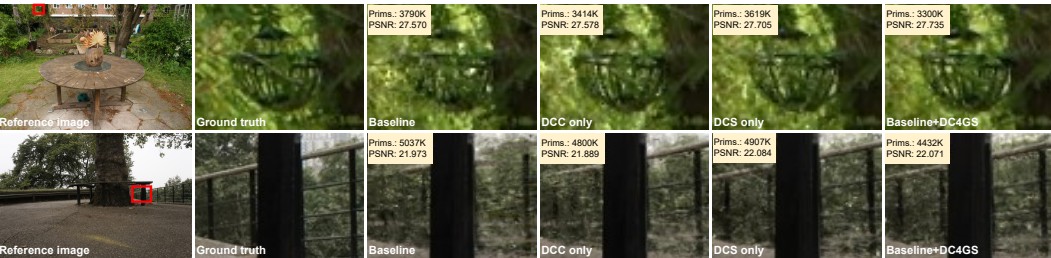

Figure 7: Qualitative visual comparison of our ablation study on the Garden and Treehill scenes from the Mip-NeRF360 dataset.

## 6.3 Ablation study

To assess the effectiveness of each component, we conduct an ablation study based on the AbsGS baseline [38]. The results are summarized in Table 3.

**Directional Consistency-weighted split Criterion (DCC).** The effect of applying the DCC (Sec.4.2) is shown in the second row of Table 3. The main effect of DCC is the reduction of the primitive counts. Here, it achieves up to a 24% reduction (Room scene). Notably, it still maintains improved or comparable quality on datasets compared to the baseline.

**Directional Consistency-guided Split (DCS).** The effect of the DCS (Sec.4.3) appears in the third row of Table 3. Its main effect is the improvement of qualities. Compared to the baseline, the DCS achieves consistently better quality even with fewer primitives.

**Combination of DCC and DCS.** Combining DCC and DCS is synergistic, which produces the most pronounced improvements. DCC and DCS individually reduce primitives and improve quality, but their integration achieves the fewest primitives and the highest fidelity. In Fig. 7, while each component alleviates artifacts such as occluded or broken cages and railings, their combination better reconstructs fine structures faithfully even with fewer primitives.

## 7 Conclusion

We introduced DC4GS, a directional consistency-driven density control for 3DGS, which selectively refines primitives by analyzing the angular coherence of positional gradients. Our density control employs directional consistency as a structural cue for both split selection and sub-primitive placement, effectively densifying primitives to align with local structural complexities. We demonstrated that our method can be integrated into the existing 3DGS pipelines and greatly enhances novel-view synthesis quality in diverse scenes with fewer primitives.

## Acknowledgement

This work was supported in part by the Mid-career Research Program and CRC Program through the NRF Grants (Nos. RS-2024-00339681 and RS-2023-00221186), and IITP grants (No. RS-2024-00454666), funded by the Korea government (MSIT). Correspondence concerning this article can be addressed to Sungkil Lee.

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

# A  Appendix

## A.1  Additional implementation details

This section provides detailed implementation components that support the overall optimization and densification process summarized in Algo. 1 of Sec. 5. Specifically, we describe the key modules in DC4GS: the evaluation of the directional consistency (DC), DC-based split costs, and the generation of sub-primitives. Each corresponds to a modified line in Algo. 1, and is detailed in the following subsections with standalone pseudocode for clarity and reproducibility.

### A.1.1  Directional consistency

---

**Algorithm 2** Directional consistency (DC) from positional gradients
$\frac{\partial L}{\partial \mu'}$: positional gradients w.r.t. projected Gaussian center $\mu'$

---

  **function** EvalDirectionalConsistency($\frac{\partial L}{\partial \mu'}$)
      $N \leftarrow 0$
      **for all** $g \in \frac{\partial L}{\partial \mu'}$ **do**
         $N \leftarrow N + 1$
         $u \leftarrow u + \frac{g}{\|g\|}$                           $\triangleright$ accumulate unit vector of positional gradient
      $C \leftarrow \frac{1}{N} u$                                       $\triangleright$ circular mean
      **return** $\|C\|$                                     $\triangleright$ DC
  **end function**

---

Algo. 2 presents the implementation of the DC defined in Section 4.1. Given a set of 2D positional gradients $\frac{\partial L}{\partial \mu'}$, where each gradient corresponds to a pixel affected by the Gaussian, we first normalize each gradient to obtain a unit vector indicating its direction. These unit vectors are accumulated and averaged to form the circular mean $C$. The directional consistency $\kappa$ is then computed as the L2 norm $\|C\|$ of this mean vector.

### A.1.2  DC-based split cost

---

**Algorithm 3** DC-based split costs for $N$ (odd) candidates along the principal axis.
$\mu$, $S$, $R$: Gaussian center, scale, rotation; $V$: view-projection matrix

---

  **function** EvalSplitCosts($\mu$, $S$, $R$, $V$, $N$, $\frac{\partial L}{\partial \mu'}$)
      $a \leftarrow \arg\max_{i \in \{x,y,z\}} S(i)$                        $\triangleright$ select principal axis
      $p_{\text{local}} \leftarrow \mathbf{e}(a)$                               $\triangleright$ basis vector along axis $a$
      $p \leftarrow R \cdot p_{\text{local}}$                                 $\triangleright$ axis in world space
      $d \leftarrow 6 \cdot S(a)$                                   $\triangleright$ diameter of a gaussian
      $x_{\text{end}}^{3D} \leftarrow \mu + 0.5d \cdot p$                      $\triangleright$ end point of the principal axis
      $x_{\text{end}}^{2D} \leftarrow \text{Project2D}(x_{\text{end}}^{3D}, V)$
      $\delta \leftarrow d/(N+1)$                                 $\triangleright$ sampling interval
      $J \leftarrow \emptyset$, $K \leftarrow \lfloor N/2 \rfloor$
      **for** $i = -K$ to $+K$ **do**
         $x_i^{3D} \leftarrow \mu + i \cdot \delta \cdot p$, $x_i^{2D} \leftarrow \text{Project2D}(x_i^{3D}, V)$
         $J \leftarrow J \cup \text{CostAt}(x_i^{2D}, x_{\text{end}}^{2D}, \frac{\partial L}{\partial \mu'})$              $\triangleright$ Alg. 4
      **return** $J$
  **end function**

---

Algo. 3 and 4 implement the DC-based cost evaluation process for the directional consistency-guided split (DCS), as described in Sec. 4.3.

Algo. 3 samples $N$ candidate split points symmetrically along the principal axis of a Gaussian, defined as the direction of maximal scale in the anisotropic matrix $S$ and transformed to world coordinates via the rotation matrix $R$. Each 3D candidate point $x^{3D}$ is projected into image space using the

**Algorithm 4** DC-based cost for a candidate split

$x_k$: split point,   $x_{\text{end}}$: principal axis endpoint,   $(x, g)$: pixel position and gradient

---

**function** CostAt($x_k$, $x_{\text{end}}$, $\frac{\partial L}{\partial \mu'}$)

    $\mathbf{v}_{\text{axis}} \leftarrow x_{\text{end}} - x_k$             ▷ 2D principal axis direction

    $\mathcal{P}_l \leftarrow []$,    $\mathcal{P}_r \leftarrow []$,    $\hat{g}_l \leftarrow 0$,    $\hat{g}_r \leftarrow 0$

    **for all** $(x, g) \in \frac{\partial L}{\partial \mu'}$ **do**

        $\mathbf{v}_{\text{pix}} \leftarrow x - x_k$, $d \leftarrow \mathbf{v}_{\text{axis}} \cdot \mathbf{v}_{\text{pix}}$        ▷ dot product to determine side of split

        **if** $d < 0$ **then**                      ▷ left of split line

            $\mathcal{P}_l \leftarrow \mathcal{P}_l \cup \{g\}, \hat{g}_l \leftarrow \hat{g}_l + |g|$

        **else**                           ▷ right of split line

            $\mathcal{P}_r \leftarrow \mathcal{P}_r \cup \{g\}, \hat{g}_r \leftarrow \hat{g}_r + |g|$

    $\kappa_l \leftarrow$ EvalDirectionalConsistency($\mathcal{P}_l$), $\kappa_r \leftarrow$ EvalDirectionalConsistency($\mathcal{P}_r$)     ▷ Alg. 2

    $J_l \leftarrow (1 - \kappa_l) \cdot \|\hat{g}_l\|, J_r \leftarrow (1 - \kappa_r) \cdot \|\hat{g}_r\|$

    **return** $J_l + J_r$

**end function**

---

view-projection matrix $V$, and the corresponding DC-based cost is computed at each $x^{2D}$ using Algo. 4.

Algo. 4 computes the cost for a single split point $x_k$ (i.e., $x^{2D}$ from Algo. 3). A split line orthogonal to the projected principal axis divides the affected pixels into left and right subsets. For each subset, the DC $\kappa$ is evaluated using Algo. 2, and the gradients $|g|$ are accumulated. The cost for each side is computed as $(1 - \kappa) \cdot \|\hat{g}\|$, and the final cost is the sum of both sides.

### A.1.3 Sub-primitive generation in DCS

---

**Algorithm 5** Splits a Gaussian into sub-primitives at $x_{\text{opt}}$

$\mu, S, R, c, o,$: center, scale, rotation, color, and opacity of a Gaussian

---

**function** SplitGaussian($x_{\text{opt}}$, $\mu$, $S$, $R$, $c$, $o$)

    $a \leftarrow \arg\max_{i \in \{x, y, z\}} S(i)$                  ▷ select principal axis

    $p_{\text{local}} \leftarrow \mathbf{e}(a), p \leftarrow R \cdot p_{\text{local}}$             ▷ axis in world space

    $d \leftarrow 6 \cdot s(a)$                       ▷ diameter of a gaussian

    $d_l \leftarrow d \cdot (1 - x_{\text{opt}}), d_r \leftarrow d \cdot x_{\text{opt}}$

    $\mu_l \leftarrow \mu - (d_l/2) \cdot p, \mu_r \leftarrow \mu + (d_r/2) \cdot p$

    $S_l \leftarrow S, S_l(a) \leftarrow S_l(a) \cdot x_{\text{opt}}$

    $S_r \leftarrow S, S_r(a) \leftarrow S_r(a) \cdot (1 - x_{\text{opt}})$

    $o_l \leftarrow o \cdot x_{\text{opt}}, o_r \leftarrow o \cdot (1 - x_{\text{opt}})$

    $G_l \leftarrow Gaussain(\mu_l, S_l, R, c, o_l), G_r \leftarrow Gaussain(\mu_r, S_r, R, c, o_r)$

    $G \leftarrow G \cup (G_l \cup G_r)$          ▷ add sub-primitives to the Gaussian set

**end function**

---

Algo. 5 implements the sub-primitive generation step after determining the optimal split position $x_{\text{opt}}$. This operation spawns two new Gaussians by dividing the original one along its principal axis.

To perform this split, the function computes the left and right extents from the center based on the optimal split position $x_{\text{opt}}$, using the principal axis direction defined during cost evaluation (see Algo. 3). The new centers $\mu_l$ and $\mu_r$ are offset from the original center $\mu$, and the corresponding scales along the principal axis are proportionally adjusted to $x_{\text{opt}}$ and $1 - x_{\text{opt}}$. Opacity values are also redistributed in the same ratio to avoid unintentionally increasing or decreasing the overall density of the original Gaussian. Each resulting sub-primitive retains the original rotation and color and is subsequently added to the primitive set.

### A.1.4 Training-time overhead analysis

In practice, DC4GS reduces the number of primitives, which accelerates the rendering time. To quantify the computational trade-offs, we provide a breakdown of DC-related overhead in Table 4.

Table 4: Breakdown of per-iteration training-time overhead introduced by DC4GS, measured on the bicycle scene of Mip-NeRF360 dataset.

| Module | Time (ms) |
|---|---|
| Directional Consistency (DC) evaluation | 0.010 |
| DC-based split cost computation | 31.020 |
| Sub-primitive generation | 16.913 |
| **Total Additional Overhead** | **47.943** |

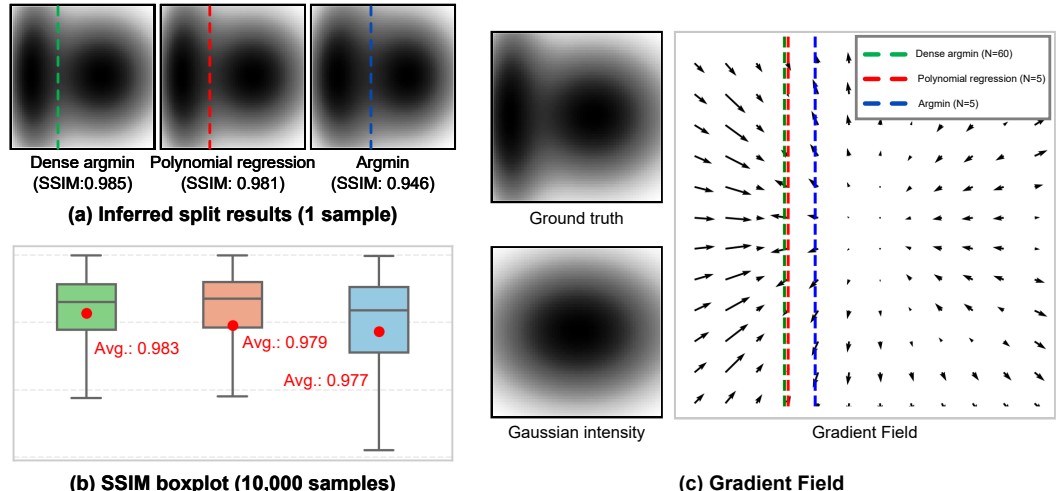

(a) Inferred split results (1 sample)

(b) SSIM boxplot (10,000 samples)

(c) Gradient Field

Figure 8: Comparison of split location selection strategies in our consistency-guided split using synthetic toy examples. (a) On a synthetic sample, we compare three strategies: dense argmin (N=60), polynomial regression (N=5), and argmin (N=5). Dense and regression methods yield sub-primitives well-aligned with the ground truth and capture structural boundaries where gradient directions conflict (see gradient field), while argmin slightly overextends the left side. (b) SSIM boxplots over 10,000 samples. Regression achieves near-equal accuracy to dense argmin. It outperforms argmin in accuracy and stability (tighter IQR), demonstrating robust performance under limited sampling.

These results are based on the Mip-NeRF360 bicycle scene with 44,206 primitives. This breakdown clarifies that most of the overhead arises from atomic operations in DC-based splitting.

## A.2 Effect of sampling resolution and selection method

We evaluate how the accuracy of split estimation is affected by the sampling resolution $N$ and the choice of selection method in our DCS. We compare three settings: 1) dense argmin ($N$=60), 2) polynomial regression [11] over $N$=5 candidates, and 3) sparse argmin ($N$=5).

Fig. 8(a) shows a synthetic example where both the dense argmin and regression identify the structural boundary well, while the sparse argmin misplaces the split, slightly overextending to the left. As shown in Fig. 8(c), this boundary aligns with a region where gradient directions conflict. The regression successfully detects this transition, preserving the underlying structure.

Fig. 8(b) shows SSIM boxplots over 10,000 samples. Dense argmin achieves the highest average (0.983), followed by the regression (0.979) and sparse argmin (0.977). The regression also exhibits the tightest distribution (median=0.987, IQR (InterQuartile Range)=0.0136), outperforming the sparse argmin (median=0.983, IQR=0.0194). These results confirm that the regression provides accurate and stable split estimation, even under limited sampling.

Table 5: Quantitative comparison for the Mip-NeRF360 scenes. From the top row to the bottom row, the results indicate PSNR, SSIM, LPIPS, and the number of primitives, respectively.

| Model | bicycle | bonsai | counter | flowers | garden | kitchen | room | stump | treehill |
|---|---|---|---|---|---|---|---|---|---|
| 3DGS [14] | 25.1331 | **32.1688** | **29.0042** | 21.4235 | 27.2435 | 31.3263 | 31.3873 | 26.5487 | **22.4933** |
| 3DGS+DC4GS | **25.1866** | 32.1594 | 28.9499 | **21.5535** | **27.3492** | **31.4530** | **31.6532** | **26.6118** | 22.4582 |
| Pixel-GS [44] | 25.2318 | **32.5194** | 29.1654 | 21.5205 | 27.3366 | 31.6594 | 31.4681 | 26.8150 | 22.1198 |
| Pixel-GS+DC4GS | **25.3140** | 32.4600 | **29.1790** | **21.7308** | **27.4599** | **31.6957** | **31.5255** | **26.8978** | **22.3223** |
| AbsGS [38] | 25.2407 | **32.3531** | **29.1104** | 21.1861 | 27.5704 | **31.8096** | **31.6676** | 26.6322 | 21.9733 |
| AbsGS+DC4GS | **25.5344** | 32.2905 | 29.0290 | **21.5621** | **27.7356** | 31.7880 | 31.6649 | **26.9562** | **22.0716** |

| Model | bicycle | bonsai | counter | flowers | garden | kitchen | room | stump | treehill |
|---|---|---|---|---|---|---|---|---|---|
| 3DGS | 0.7602 | **0.9400** | **0.9061** | 0.6019 | 0.8619 | 0.9259 | 0.9178 | 0.7691 | 0.6307 |
| 3DGS+DC4GS | **0.7655** | 0.9399 | 0.9060 | **0.6059** | **0.8638** | **0.9261** | **0.9188** | **0.7743** | **0.6331** |
| Pixel-GS | 0.7757 | **0.9445** | 0.9121 | 0.6337 | 0.8661 | 0.9292 | 0.9206 | 0.7836 | 0.6316 |
| Pixel-GS+DC4GS | **0.7798** | 0.9442 | **0.9122** | **0.6403** | **0.8680** | **0.9293** | **0.9212** | **0.7876** | **0.6375** |
| AbsGS | 0.7800 | **0.9453** | **0.9120** | 0.6174 | 0.8713 | 0.9297 | **0.9253** | 0.7753 | 0.6133 |
| AbsGS+DC4GS | **0.7925** | 0.9445 | 0.9113 | **0.6348** | **0.8741** | **0.9300** | 0.9252 | **0.7924** | **0.6287** |

| Model | bicycle | bonsai | counter | flowers | garden | kitchen | room | stump | treehill |
|---|---|---|---|---|---|---|---|---|---|
| 3DGS | 0.2159 | **0.2050** | **0.2019** | 0.3411 | 0.1092 | **0.1265** | 0.2197 | 0.2186 | **0.3292** |
| 3DGS+DC4GS | **0.2104** | 0.2058 | 0.2026 | **0.3366** | **0.1083** | 0.1267 | **0.2189** | **0.2142** | 0.3310 |
| Pixel-GS | 0.1818 | **0.1934** | **0.1844** | **0.2617** | 0.1003 | **0.1195** | 0.2110 | 0.1868 | **0.2787** |
| Pixel-GS+DC4GS | **0.1793** | 0.1938 | 0.1857 | 0.2654 | **0.0996** | 0.1199 | **0.2105** | **0.1858** | 0.2848 |
| AbsGS | 0.1729 | **0.1894** | **0.1874** | 0.2713 | 0.0992 | **0.1206** | **0.1999** | 0.1979 | 0.2801 |
| AbsGS+DC4GS | **0.1663** | 0.1907 | 0.1892 | **0.2684** | **0.0977** | 0.1211 | 0.2018 | **0.1884** | **0.2733** |

| Model | bicycle | bonsai | counter | flowers | garden | kitchen | room | stump | treehill |
|---|---|---|---|---|---|---|---|---|---|
| 3DGS | 6056K | 1285K | 1226K | 3618K | 5694K | 1839K | 1592K | 4895K | 3944K |
| 3DGS+DC4GS | **5299K** | **1124K** | **1112K** | **3389K** | **5269K** | **1677K** | **1419K** | **3980K** | **3438K** |
| Pixel-GS | 9028K | 2102K | 2603K | 7517K | 8574K | 3180K | 2588K | 6610K | 8396K |
| Pixel-GS+DC4GS | **8047K** | **1860K** | **2378K** | **6966K** | **7914K** | **2945K** | **2332K** | **5730K** | **6906K** |
| AbsGS | 6051K | 1045K | 970K | 3886K | 3790K | 1243K | 1471K | 4845K | 5037K |
| AbsGS+DC4GS | **5021K** | **830K** | **754K** | **3330K** | **3300K** | **930K** | **1036K** | **3898K** | **4432K** |

Table 6: Quantitative comparison for the Tanks&Temples scenes. From the top row to the bottom row, the results indicate PSNR, SSIM, LPIPS, and the number of primitives (Prim.), respectively.

| Method | Train | | | | Truck | | | |
|---|---|---|---|---|---|---|---|---|
| | PSNR ↑ | SSIM ↑ | LPIPS ↓ | Prim. | PSNR ↑ | SSIM ↑ | LPIPS ↓ | Prim. |
| 3DGS | 21.8415 | 0.8103 | **0.2103** | 1099K | 25.4690 | 0.8777 | 0.1486 | 2687K |
| 3DGS+DC4GS | **21.9974** | **0.8116** | 0.2112 | **1048K** | **25.5747** | **0.8801** | **0.1469** | **2358K** |
| Pixel-GS | 21.9925 | 0.8236 | 0.1804 | 3857K | 25.5262 | 0.8828 | 0.1216 | 5338K |
| Pixel-GS+DC4GS | **22.2010** | **0.8260** | **0.1799** | **3561K** | **25.6590** | **0.8856** | **0.1213** | **4650K** |
| AbsGS | 21.5800 | 0.8178 | 0.1927 | 1008K | 25.6926 | 0.8869 | 0.1322 | 1657K |
| AbsGS+DC4GS | **22.3767** | **0.8291** | **0.1871** | **787K** | **25.8666** | **0.8894** | **0.1317** | **1399K** |

Table 7: Quantitative comparison for the Deep Blending scenes. From the top row to the bottom row, the results indicate PSNR, SSIM, LPIPS, and the number of primitives (Prim.), respectively.

| Method | Dr Johnson | | | | Playroom | | | |
|---|---|---|---|---|---|---|---|---|
| | PSNR ↑ | SSIM ↑ | LPIPS ↓ | Prim. | PSNR ↑ | SSIM ↑ | LPIPS ↓ | Prim. |
| 3DGS | 29.0423 | 0.8977 | 0.2471 | 3321K | 29.7464 | 0.8997 | 0.2490 | 2345K |
| 3DGS+DC4GS | **29.1892** | **0.9000** | **0.2455** | **3158K** | **29.9417** | **0.9021** | **0.2459** | **2131K** |
| Pixel-GS | 27.9112 | 0.8843 | 0.2602 | 5491K | 29.7131 | 0.8989 | 0.2446 | 3754K |
| Pixel-GS+DC4GS | **28.5347** | **0.8914** | **0.2500** | **5203K** | **29.8276** | **0.9010** | **0.2429** | **3365K** |
| AbsGS | 29.0475 | 0.8956 | 0.2420 | 2455K | 29.9527 | 0.9049 | **0.2330** | 1467K |
| AbsGS+DC4GS | **29.2328** | **0.9027** | **0.2370** | **1905K** | **30.0759** | **0.9087** | 0.2331 | **1093K** |

### A.3 Additional experimental results

#### A.3.1 Per-scene quantitative results

Tables 5–7 report detailed per-scene metrics, including PSNR, SSIM, LPIPS, and the number of primitives, for all datasets. The results show that DC4GS consistently reduces the number of primitives and maintains or improves reconstruction quality compared to each baseline method.

In the stump scene, applying DC4GS to 3DGS results in up to a 19% reduction in primitives. With Pixel-GS [44], DC4GS achieves up to an 18% reduction in the treehill scene. The largest reduction is observed with AbsGS [38], where DC4GS reduces the number of primitives by up to 30% in the room scene. In all cases, PSNR, SSIM, and LPIPS metrics remain superior or comparable to each baseline, demonstrating that DC4GS enhances both compactness and reconstruction fidelity in 3DGS-based pipelines.

#### A.3.2 Additional qualitative comparisons

Fig. 9–11 provide additional qualitative comparisons between the baselines (Pixel-GS [44], 3DGS [14], and AbsGS [38]) and their DC4GS-integrated models.

**Comparisons with Pixel-GS** As shown in Fig. 9, DC4GS more effectively preserves linear boundaries, such as door panel patterns, counter edges, window frames, and lane markings. It also improves contrast in fine details and maintains distinct separation between adjacent regions, mitigating overshooting and bleeding artifacts.

**Comparisons with 3DGS** Fig. 10 shows that DC4GS improves the sharpness of architectural lines and geometric boundaries. It more accurately reconstructs fine structures like door panels, counter surfaces, window outlines, and wall trims. Straight edges and corners appear cleaner and less distorted, with reduced blending between adjacent objects and surfaces.

**Comparisons with AbsGS** In Fig. 11, DC4GS more reliably recovers structures such as window frames, rods, and wall seams, and reduces floating artifacts and blending at region boundaries (e.g., between sky, trees, and train). It suppresses redundant splits and improves local separation, resulting in clearer reconstructions with sharper object boundaries and fewer visual artifacts.

DC4GS consistently improves the reconstruction of fine structures and local detail, enhancing the separation between distinct objects and surfaces. This leads to reduced structural blending and clearer geometry in various scenes.

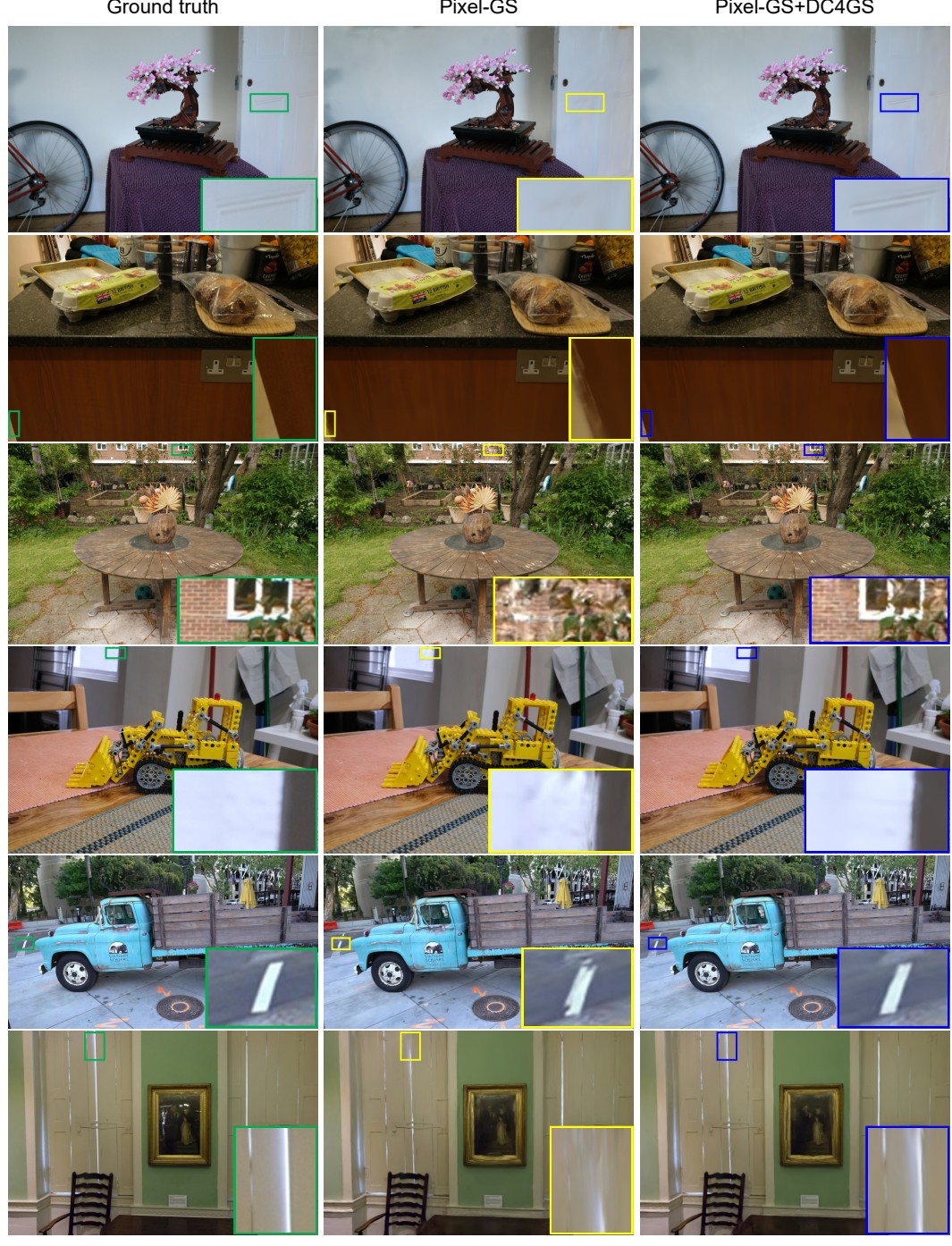

Figure 9: Qualitative visual comparisons between PixelGS and PixelGS integrated with DC4GS. The scenes, from top to bottom, are: Bonsai, Counter, Garden, Kitchen from the Mip-NeRF360 dataset, Truck from the Tanks&Temples dataset, and Dr Johnson from the Deep Blending dataset.

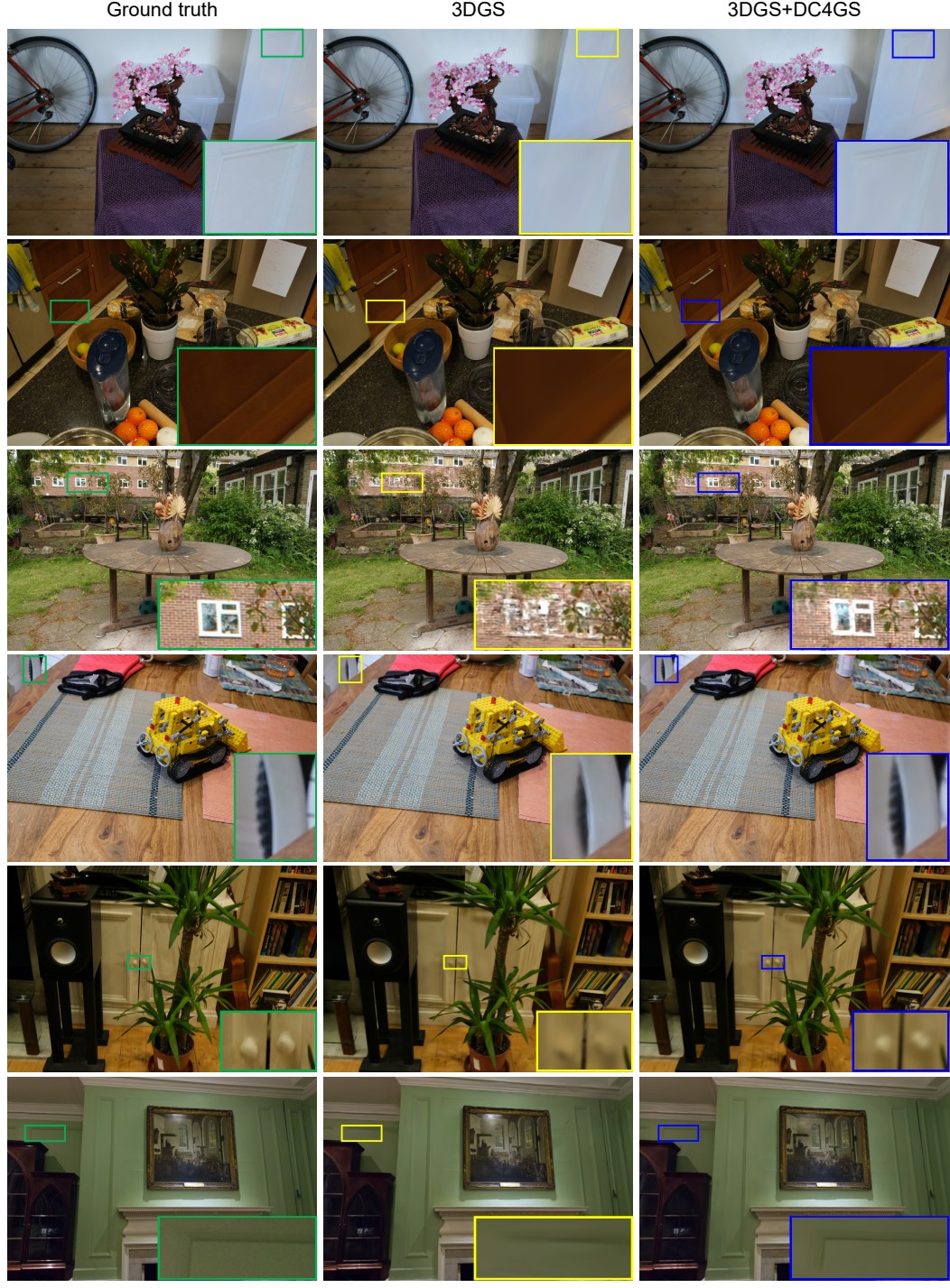

Figure 10: Qualitative visual comparisons between 3DGS and 3DGS integrated with DC4GS. The scenes, from top to bottom, are: Bonsai, Counter, Garden, Kitchen, Room from the Mip-NeRF360 dataset, and Dr Johnson from the Deep Blending dataset.

Ground truth       AbsGS       AbsGS+DC4GS

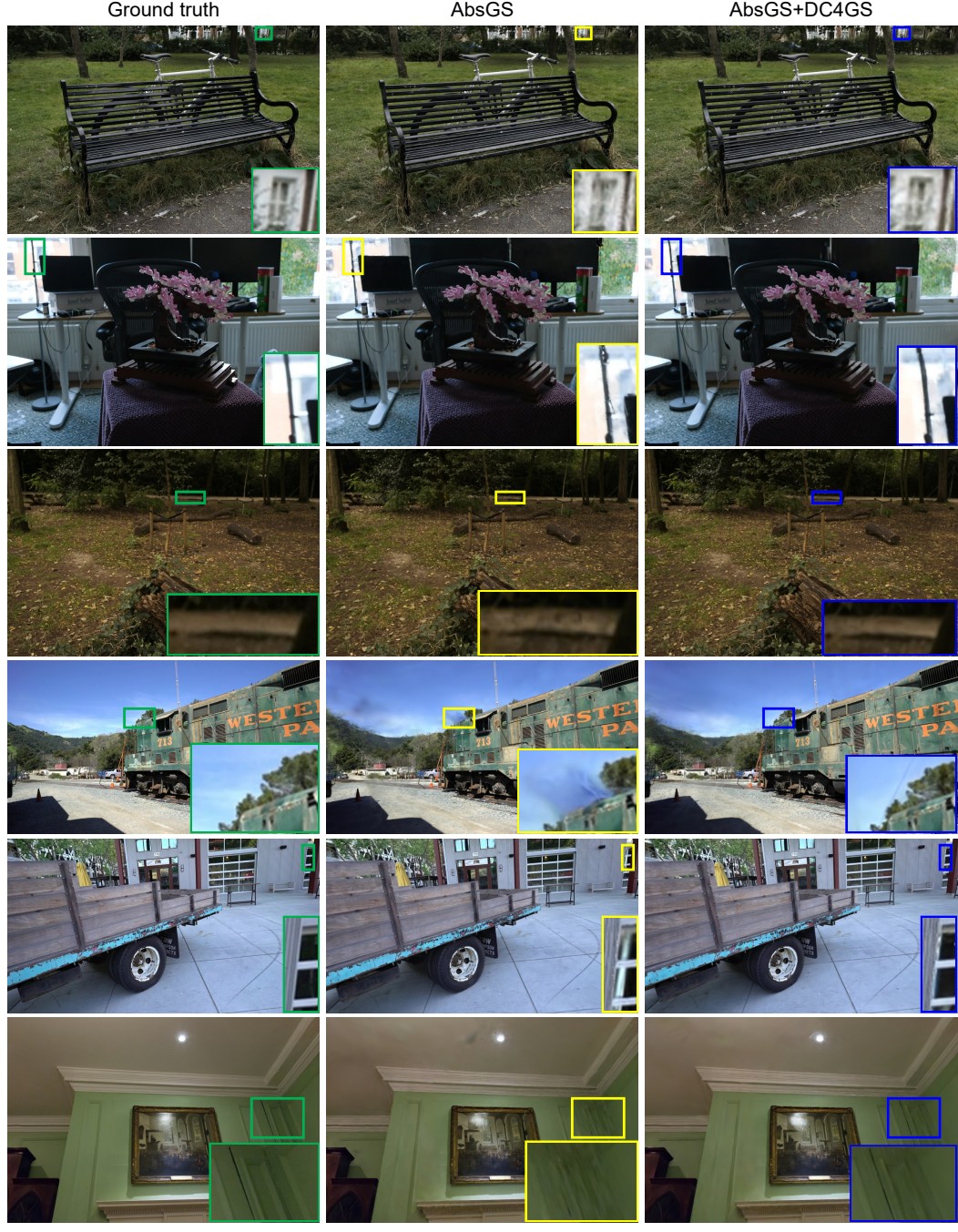

Figure 11: Qualitative visual comparisons between AbsGS and AbsGS integrated with DC4GS. The scenes, from top to bottom, are: Bicycle, Bonsai, Stump from the Mip-NeRF360 dataset, Train, Truck from the Tanks&Temples dataset, and Dr Johnson from the Deep Blending dataset.

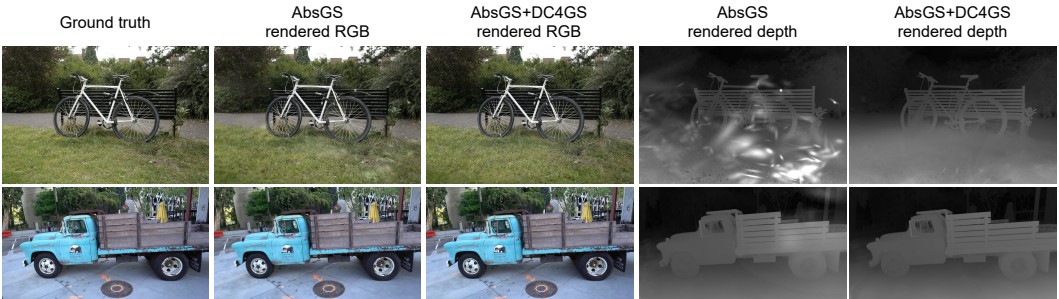

Figure 12: Qualitative depth visualization comparison between AbsGS and AbsGS integrated with DC4GS. The scenes, from top to bottom, are: Bicycle from the Mip-NeRF360 dataset, and Truck from the Tanks&Temples dataset.

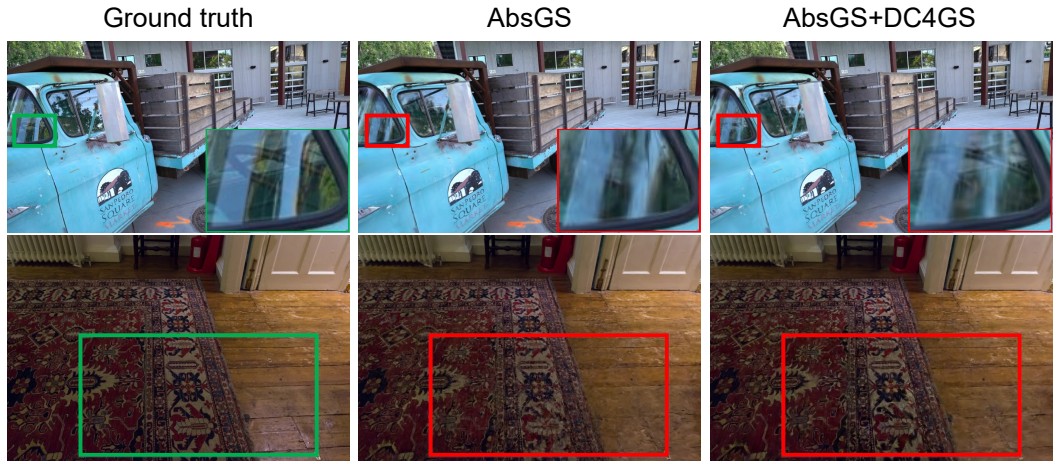

Figure 13: Qualitative comparisons on challenging cases between AbsGS and AbsGS integrated with DC4GS. The scenes, from top to bottom, are: Truck from the Tanks&Temples dataset (transparent objects), and Dr Johnson from the Deep Blending dataset (strong textures).

Table 8: Quantitative comparison between AbsGS and AbsGS integrated with DC4GS on challenging cases: transparency (Truck in Tanks&Temples) and strong textures (Dr. Johnson in Deep Blending).

| Scene | Region | Method | PSNR ↑ | SSIM ↑ | LPIPS ↓ |
|---|---|---|---|---|---|
| Truck | Transparent object | AbsGS | 20.875 | 0.621 | 0.333 |
| | | AbsGS + DC4GS | **21.098** | **0.642** | **0.301** |
| Dr. Johnson | Strong texture | AbsGS | 25.741 | 0.717 | 0.334 |
| | | AbsGS + DC4GS | **26.462** | **0.761** | **0.294** |

### A.3.3   Depth visualization comparison

We qualitatively assess geometric fidelity through depth map visualizations. To obtain depth maps, we adapt the original 3DGS rendering equation by replacing the RGB color contribution with per-Gaussian depth values $d_i$:

$$D = \sum_i T_i \cdot \alpha_i \cdot d_i, \quad T_i = \prod_{j=1}^{i-1}(1 - \alpha_j), \tag{4}$$

where $\alpha_i$ is the alpha and $T_i$ is the transmittance value of the $i$-th Gaussian primitive.

As shown in Fig. 12, The baseline produces noisy depth maps. This noise mainly stems from the presence of numerous false-positive primitives that persist without proper density control. Such

Table 9: Quantitative comparison of 4DGS and 4DGS with DC4GS on D-NeRF and DyNeRF datasets, reporting quality metrics (PSNR, SSIM, LPIPS) and efficiency metrics (number of primitives, memory, training time, and rendering time in ms).

| Dataset | Method | PSNR ↑ | SSIM ↑ | LPIPS ↓ | Prim. ↓ | Mem. ↓ | Training ↓ | Rendering ↓ |
|---------|--------|--------|--------|---------|---------|--------|------------|-------------|
| D-NeRF [27] | 4DGS [36] | 34.063 | **0.978** | **0.026** | 45K | 10MB | **9m** | 15.233ms |
|  | 4DGS + DC4GS | **34.124** | **0.978** | **0.026** | **40K** | **9MB** | 11m | **14.531ms** |
| DyNeRF [20] | 4DGS | 30.736 | 0.933 | 0.059 | 123K | 59MB | **50m** | 32.537ms |
|  | 4DGS + DC4GS | **31.092** | **0.936** | **0.056** | **119K** | **56MB** | 1h 13m | **32.259ms** |

primitives contribute spurious depth values, leading to fragmented and unstable geometry. In contrast, DC4GS effectively suppresses false-positive primitives, yielding cleaner and more coherent depth reconstructions. This suggests that the benefits of DC4GS extend beyond RGB fidelity to improved structural consistency.

### A.3.4 Applicability to challenging scenarios

To evaluate DC4GS under challenging conditions, we perform region-specific comparisons on transparency and strong textures. Quantitative results, measured on the masked inset regions in Fig. 13, are reported in Table 8, and the corresponding qualitative comparisons are also shown in Fig. 13.

For the transparent objects (4th test image of the Truck scene in Tanks & Temples), the baseline fails to reconstruct the internal features behind the glass (e.g., steering wheel), whereas DC4GS successfully preserves these structures. For strong high-frequency textures (9th test image of the Dr. Johnson scene in Deep Blending), such as carpet patterns and wood scratches, DC4GS achieves superior metrics over the baseline, showing its ability to capture fine details. These results suggest that DC4GS can better handle transparent and high-frequency regions that are challenging for the baseline.

### A.3.5 Applicability to dynamic scenes

We further evaluate DC4GS on dynamic scenes by integrating it into 4DGS [36] and testing on D-NeRF [27] and DyNeRF [20]. As shown in Table 9, DC4GS yields consistent improvements in reconstruction quality, while reducing the number of primitives and memory usage, which also leads to faster rendering. Qualitative comparisons in Fig. 14 show that DC4GS produces sharper geometry and more temporally stable appearance, whereas 4DGS often exhibits blurred details and temporal artifacts. Despite the increased training time, these results suggest that DC4GS remains effective and robust in dynamic scenarios.

### A.4 Limitations and future work

DC4GS enables high-quality reconstruction with substantially fewer primitives, resulting in faster rendering and reduced memory consumption for 3DGS models. Although the method introduces additional training overhead due to directional consistency evaluation and candidate split cost estimation, this overhead is confined to the training phase and can be entirely removed once the densification stage is complete (typically after 15,000 iterations). Future work will focus on optimizing the DC computation and split estimation pipeline, particularly the cost evaluation and polynomial regression, to further reduce training time without compromising reconstruction quality.

### A.5 Broader impact

DC4GS improves the efficiency of 3DGS by reducing primitive count without compromising quality. This leads to lower memory usage for 3DGS parameters and faster rendering, making high-quality 3D reconstruction more accessible for real-time and resource-constrained applications such as AR/VR and mobile platforms.

As with other 3D reconstruction techniques, there is potential for misuse (e.g., unauthorized duplication of real-world scenes). Responsible use and ethical considerations remain important in downstream applications.

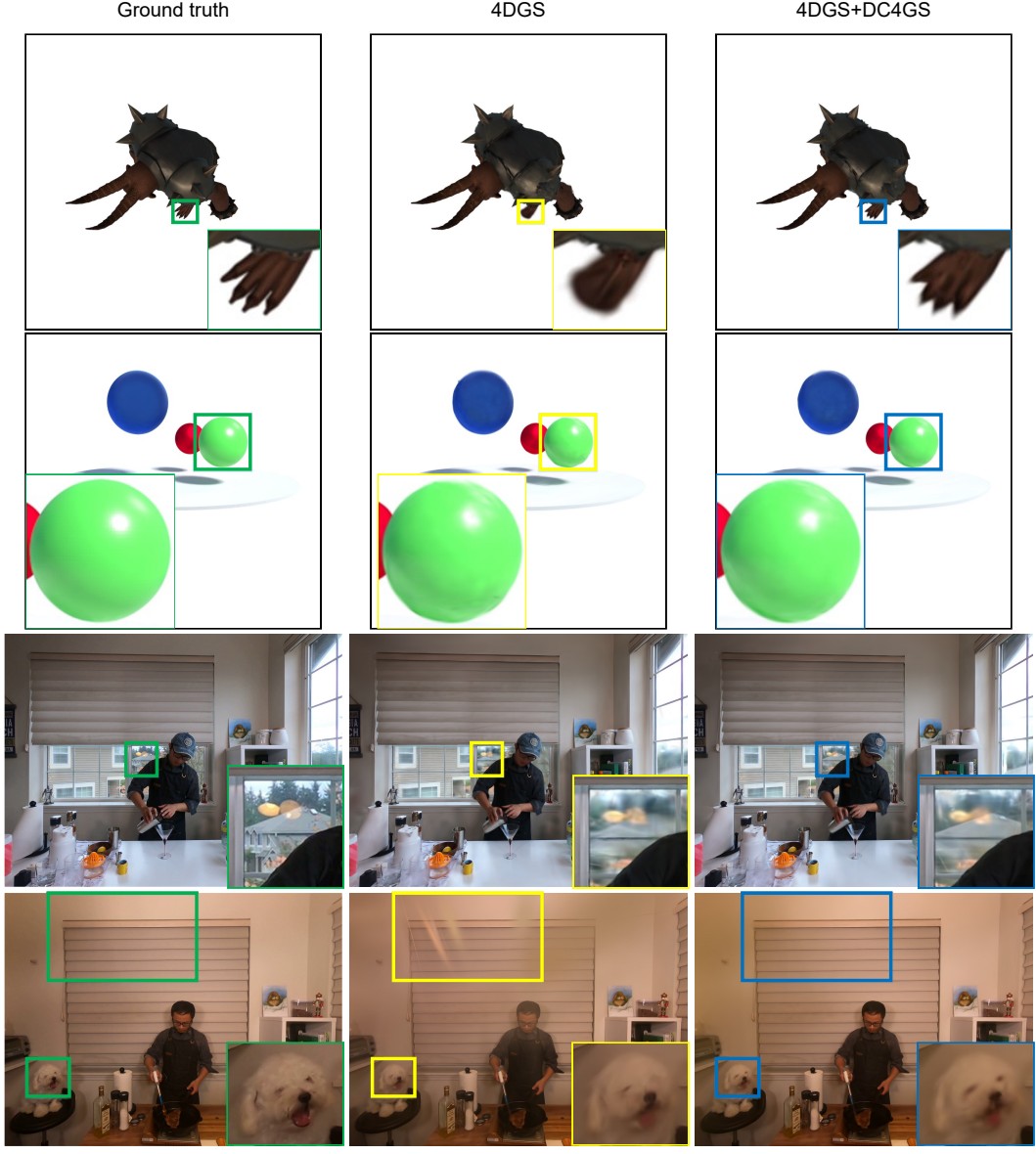

Figure 14: Qualitative comparisons on dynamic scenes between 4DGS and 4DGS with DC4GS. The scenes, from top to bottom, are: Bouncing balls and Hell warrior from the D-NeRF dataset, and Coffee martini and Flame steak from the DyNeRF dataset.

## A.6 Dataset licenses

We use the following datasets in our experiments:

- Mip-NeRF360 [2]: no explicit license terms provided. Available at https://jonbarron.info/mipnerf360/.
- Tanks and Temples [16]: released under the Creative Commons Attribution 4.0 International (CC BY 4.0). Available at https://www.tanksandtemples.org/license/.
- Deep Blending [12]: no explicit license terms provided. Available at http://visual.cs.ucl.ac.uk/pubs/deepblending/.

