# OpenReview forum: "DC4GS: Directional Consistency-Driven Adaptive Density Control for 3D Gaussian Splatting"
_NeurIPS.cc/2025/Conference — NeurIPS 2025 poster_

### Official Review · Reviewer_1vr6 · 2025-06-17

**Clarity:** 3
**Significance:** 2
**Originality:** 3
**Rating:** 3
**Confidence:** 4

**Summary:**

This paper handles the problem of the gradient direction being ignored in original ADC of 3DGS. The authors use directional consistency to assess the directional coherence of its positional gradients. The directional consistency are used to choose the gradient indicating that a Gaussian needs to be split. To make the sub-primitives better fit the regions, the polynomial regression is used to find the optimal split position under min DC-based cost. The experiments show the proposed methods obtain a better rendering quality with less Gaussians.

**Questions:**

+ I am not very sure that Line 145 and 195 should be "larger than the scale threshold".
+ I would like to know the value of thresholds $\tau_p$ and $\tau_S$ in Algorithm 1.
+ Could the sub-primitives from the splitting operation in the original ADC of 3DGS adjust their positions in the subsequent training process to finally fit the target regions? I think that might be the reason why the improvements caused by the proposed splitting strategy and other methods are not obvious.

**Ethical Concerns:**

["NO or VERY MINOR ethics concerns only"]

**Final Justification:**

My main concerns about the performance of the proposed methods have not been addressed. Although this paper has a clear motivation, the experiments on most datasets have unsatisfactory improvements. Only performing well on the ``Dr Johnson'' scene cannot demonstrate the effectiveness of the proposed methods. Also, the geometry evaluation based on Depth Anything cannot convince me because the pseudo GTs generated by DPT are not accurate. In general, I decide to maintain my rating as boardline reject.

**Limitations:**

Yes, the limitation has been mentioned in the supplementary.

**Paper Formatting Concerns:**

No formatting concerns.

**Quality:**

3

**Strengths And Weaknesses:**

**Strengths**

+ The direction of gradient is useful in distinguishing Gaussians required to be split. The proposed methods are well designed with sufficient theoretical support.
+ The proposed methods seem novel, and the experiments show a certain improvement effect.

**Weakness**

+ The improvements shown in the experiments are not obvious, which probably cannot demonstrate the effectiveness of the proposed methods.
+ It seems that there is the same problem in the ablation study.

---

> ### Author Rebuttal · Authors · 2025-07-31
>
> ## Clarification on the Effectiveness of DC4GS
>
> >**Weakness 1**
> > "The improvements shown in the experiments are not obvious, which probably cannot demonstrate the effectiveness of the proposed methods."
>
> We appreciate the reviewer’s comment regarding the seemingly modest improvements in numerical metrics. While the absolute gains in PSNR, SSIM, and LPIPS may appear limited, the main contribution of DC4GS lies in its ability to achieve a superior trade-off between memory efficiency and rendering quality.
> Specifically, DC4GS consistently reduces the number of primitives and memory usage by 10–30%, achieving comparable or superior rendering quality. This is further supported by its compatibility with a variety of 3DGS architectures.
>
> To substantiate DC4GS effectiveness, we additionally integrated it into several recent 3DGS pipelines, including Scaffold-GS [1], GES [2], and 4DGS [3]. For Scaffold-GS + DC4GS, since Scaffold-GS does not include split operation, we apply only the Directional Consistency-weighted split Criterion (DCC) to modulate anchor growth.
> The observed improvements in each case demonstrate the practicality and generality of our method. Notably, on the DyNeRF [5] dataset using 4DGS, a method tailored for dynamic scene reconstruction, DC4GS not only reduces memory usage but also yields a **1.2 dB** gain in PSNR.
> This result is particularly meaningful, as it shows our method is not confined to static scenarios but also effective in dynamic and temporally varying settings, thereby reinforcing its general applicability and practical utility beyond mere metric gains.
>
> > ### D-NeRF and DyNeRF dataset
> |Dataset|Method|PSNR ↑|SSIM ↑|LPIPS ↓|Prim ↓|Mem ↓|Training Time ↓|Rendering Time ↓|
> |-|-|-|-|-|-|-|-|-|
> |D-NeRF [4]|4DGS|34.063|**0.978**|**0.026**|45K|10MB|**9m**|15.233ms|
> ||4DGS + DC4GS |**34.124**|**0.978**|**0.026**|**40K**|**9MB**|11m|**14.531ms**|
> |DyNeRF [5]|4DGS|29.718|0.930|0.153|123K|29MB|**50m**|**32.137ms**|
> ||4DGS + DC4GS|**30.983**|**0.937**|**0.148**|**118K**|**28MB**|1h 13m|32.275ms|
>
> > ### Mip-NeRF360 dataset
> |Method|PSNR ↑|SSIM ↑|LPIPS ↓|Prim ↓|Mem ↓|Training Time ↓|Rendering Time ↓|
> |-|-|-|-|-|-|-|-|
> |Scaffold-GS [1]|**27.578**|0.809|0.225|569K|171MB|**49m**|9.244ms|
> |Scaffold-GS + DC4GS|27.577|**0.81**|0.225|**425K**|**128MB**|50m|**8.537ms**|
> |GES [2]|27.04|0.796|0.248|1543K|365MB|**32m**|6.981ms|
> |GES + DC4GS|**27.09**|**0.797**|0.248|**1323K**|**313MB**|42m|**6.698ms**|
>
> > ### Tanks&Temples dataset
> |Method|PSNR ↑|SSIM ↑|LPIPS ↓|Prim ↓|Mem ↓|Training Time ↓|Rendering Time ↓|
> |-|-|-|-|-|-|-|-|
> |Scaffold-GS [1]|**24.018**|**0.851**|**0.174**|247K|74MB|**29m**|7.015ms|
> |Scaffold-GS + DC4GS|24.016|0.849|0.177|**208K**|**62MB**|31m|**6.933ms**|
> |GES [2]| **23.64** |**0.842**|**0.191**|930K|220MB|**18m**|5.593ms|
> |GES + DC4GS|23.515|0.841|0.194|**816K**|**193MB**|22m|**5.441ms**|
>
> > ### Deep Blending dataset
> |Method|PSNR ↑|SSIM ↑|LPIPS ↓|Prim ↓|Mem ↓|Training Time ↓|Rendering Time ↓|
> |-|-|-|-|-|-|-|-|
> |Scaffold-GS [1]|**30.252**|**0.904**|**0.255**|171K|51MB|**42m**|6.298ms|
> |Scaffold-GS + DC4GS|30.063|0.903|0.257|**135K**|**40MB**|44m|**5.748ms**|
> |GES [2]|29.562|0.903|0.249|1606K|380MB|**40m**|6.835ms|
> |GES + DC4GS|**29.632**|**0.904**|**0.248**|**1487K**|**352MB**|45m|**6.571ms**|
>
> ## On the Effectiveness of the DC-driven ADC vs. Original ADC
>
> > **Questions 3**
> > "Could the sub-primitives from the splitting operation in the original ADC of 3DGS adjust their positions in the subsequent training process to finally fit the target regions? I think that might be the reason why the improvements caused by the proposed splitting strategy and other methods are not obvious."
>
> We appreciate the reviewer’s insightful question. To evaluate whether the sub-primitives generated by the original ADC in 3DGS can eventually adapt to their target regions through prolonged optimization, we conducted additional experiments using AbsGS as the baseline. Specifically, we doubled the training iterations from the default 30,000 to 60,000, allowing ample time for the sub-primitives to adjust their positions if possible.
>
> > ### Mip-NeRF360 dataset
> |Method|PSNR ↑|SSIM ↑|LPIPS ↓|Prim ↓|Mem ↓|Training Time ↓|Rendering Time ↓|
> |-|-|-|-|-|-|-|-|
> |AbsGS|27.504|0.818|0.191|3149K|745MB|**37m**|9.975ms|
> |AbsGS (60K iter.)|27.539|0.817|0.189|3162K|748MB|1h 19m|10.247ms|
> |AbsGS + DC4GS|**27.625**|**0.826**|**0.188**|**2615K**|**618MB**|1h 4m|**9.225ms**|
>
> > ### Tanks&Temples dataset
> |Method|PSNR ↑|SSIM ↑|LPIPS ↓|Prim ↓|Mem ↓|Training Time ↓|Rendering Time ↓|
> |-|-|-|-|-|-|-|-|
> |AbsGS|23.636|0.852|0.162|1332K|315MB|**17m**|6.175ms|
> |AbsGS (60K iter.)|24.039|0.855|**0.156**|1313K|311MB|38m|6.69ms|
> |AbsGS + DC4GS|**24.121**|**0.859**|0.159|**1093K**|**259MB**|28m|**5.887ms**|
>
> > ### Deep Blending dataset
> |Method|PSNR ↑|SSIM ↑|LPIPS ↓|Prim ↓|Mem ↓|Training Time ↓|Rendering Time ↓|
> |-|-|-|-|-|-|-|-|
> |AbsGS|29.500|0.900|0.237|1961K|464MB|**27m**|7.353ms|
> |AbsGS (60K iter.)|29.275|0.895|0.240|1962K|464MB|57m|7.549ms|
> |AbsGS + DC4GS|**29.654**|**0.905**|**0.235**|**1499K**|**354MB**|50m|**6.627ms**|
>
> Despite this extended training, DC4GS still outperforms the baseline using the original ADC of 3DGS, both in rendering quality and structural fidelity. To further investigate whether prolonged optimization could compensate for suboptimal primitive splits, we trained the AbsGS baseline with original ADC for 60K iterations—twice the default schedule.
> While this extension yielded marginal improvements on the Mip-NeRF360 and Tanks&Temples datasets, it resulted in a performance drop on the Deep Blending dataset. This outcome suggests that simply increasing the number of training iterations is insufficient to correct suboptimal initial splits, as downstream optimization alone cannot fully recover from poor primitive placement.
>
> In contrast, our Directional Consistency (DC)-guided ADC ensures that sub-primitives are initialized in geometrically meaningful and directionally coherent locations at the moment of splitting. This guided placement accelerates convergence and leads to superior reconstruction performance.
> These results demonstrate that directionally guided initialization is crucial for effective convergence, emphasizing the importance of incorporating DC in the ADC design.
>
> ## Ablation Study Effectiveness
>
> > **Weakness 2**
> > "It seems that there is the same problem in the ablation study."
>
> We appreciate the reviewer’s observation regarding the relatively modest impact of each component in the ablation study. While each module may contribute only limited improvement, they are designed to address distinct aspects of the task. When combined, they yield a greater overall improvement, both qualitatively and quantitatively, demonstrating their complementary roles in addressing different aspects of the task. Notably, as reported in the paper, the full model achieves enhanced rendering quality, reducing the number of primitives by approximately 20% on average.
>
> ## Clarification on Lines 145/195 and Threshold Values in Algorithm 1
>
> > **Questions 1 and 2**
> > "I am not very sure that Line 145 and 195 should be 'larger than the scale threshold'."
> > "I would like to know the value of thresholds $ \tau_p $ and $ \tau_S $ in Algorithm 1."
>
> We thank the reviewer for raising these helpful questions that help improve the clarity of our paper.
>
> Regarding the first question: the phrasing  "larger than the scale threshold" in Lines 145 and 195 follows the standard densification scheme used in 3D Gaussian Splatting (3DGS).
> As described in the Preliminary section, a Gaussian $i$ is split when both of the following conditions are satisfied:
>
> (1) the magnitude of the positional gradient $ \|\nabla_{\mu_i'} \mathcal{L}\| > \tau_p $
> (2) the maximum scale $ \|S_i\| < \tau_S $
>
> In our method, we preserve second condition (scale threshold) and only modify the standard gradient-based criterion $ \nabla_{\mu_i'} \mathcal{L} $ by introducing the DCC, denoted as $ \nabla_{{\mu'}_i}^{DC} \mathcal{L} $  in the paper.
>
> That is, we replace the original positional gradient magnitude $ \|\nabla_{\mu_i'} \mathcal{L}\| $ with our DCC, while keeping the same scale-based condition $ \|S_i\| < \tau_S $.
>
> For the second question: we adopt the default threshold values $\tau_p$ and $\tau_S$ as defined in each baseline method without modification. For clarity, we list the values used in our experiments:
>
> **3DGS**: $ \tau_p = 0.0002, \tau_S = 0.01 \times $ scene_extent
> **AbsGS**: $ \tau_p = 0.0004, \tau_S = 0.001 \times $ scene_extent
> **PixelGS**: $ \tau_p = 0.0002, \tau_S = 0.01 \times $ scene_extent
>
> Here, scene_extent is not a new parameter, but rather a dataset-specific normalization scalar already defined and used in the original 3DGS. It reflects the geometric extent of the scene as observed from the distribution of training camera poses.
>
> We will clarify these details explicitly in the revised manuscript to avoid confusion.
>
>
> ## References
>
> [1] Lu, Tao, et al. "Scaffold-GS: Structured 3d gaussians for view-adaptive rendering." Proceedings of the IEEE/CVF Conference on Computer Vision and Pattern Recognition. 2024.
> [2] Hamdi, Abdullah, et al. "GES: Generalized exponential splatting for efficient radiance field rendering." Proceedings of the IEEE/CVF Conference on Computer Vision and Pattern Recognition. 2024.
> [3] Wu, Guanjun, et al. "4D gaussian splatting for real-time dynamic scene rendering." Proceedings of the IEEE/CVF conference on computer vision and pattern recognition. 2024.
> [4] Pumarola, Albert, et al. "D-NeRF: Neural radiance fields for dynamic scenes." Proceedings of the IEEE/CVF conference on computer vision and pattern recognition. 2021.
> [5] Li, Tianye, et al. "Neural 3d video synthesis from multi-view video." Proceedings of the IEEE/CVF conference on computer vision and pattern recognition. 2022.

---

> > ### Author Response · Authors · 2025-08-01
> > **Correction to Reported Training Times in the Table of the Rebuttal**
> >
> > ## On the Effectiveness of the DC-driven ADC vs. Original ADC (Training Time Correction)
> >
> > > **Questions 3**
> > > "Could the sub-primitives from the splitting operation in the original ADC of 3DGS adjust their positions in the subsequent training process to finally fit the target regions? I think that might be the reason why the improvements caused by the proposed splitting strategy and other methods are not obvious."
> >
> > We sincerely apologize for the mistake in the reported training time of `AbsGS + DC4GS` in our rebuttal, which included extra time caused by logging operations. The correct training times are 51 minutes, 23 minutes, and 40 minutes for the Mip-NeRF360, Tanks&Temples, and Deep Blending datasets, respectively, as correctly listed in Table 6 of the submitted Appendix.
> > We have updated the corresponding table below to reflect these corrections, and kindly ask the reviewer to refer to the revised version for accurate comparison. We sincerely apologize for any confusion this may have caused and thank the reviewer for their understanding.
> >
> > > ### Mip-NeRF360 dataset
> > |Method|PSNR ↑|SSIM ↑|LPIPS ↓|Prim ↓|Mem ↓|Training Time ↓|Rendering Time ↓|
> > |-|-|-|-|-|-|-|-|
> > |AbsGS|27.504|0.818|0.191|3149K|745MB|**37m**|9.975ms|
> > |AbsGS (60K iter.)|27.539|0.817|0.189|3162K|748MB|1h 19m|10.247ms|
> > |AbsGS + DC4GS|**27.625**|**0.826**|**0.188**|**2615K**|**618MB**|51m ~~1h 4m~~|**9.225ms**|
> >
> > > ### Tanks&Temples dataset
> > |Method|PSNR ↑|SSIM ↑|LPIPS ↓|Prim ↓|Mem ↓|Training Time ↓|Rendering Time ↓|
> > |-|-|-|-|-|-|-|-|
> > |AbsGS|23.636|0.852|0.162|1332K|315MB|**17m**|6.175ms|
> > |AbsGS (60K iter.)|24.039|0.855|**0.156**|1313K|311MB|38m|6.69ms|
> > |AbsGS + DC4GS|**24.121**|**0.859**|0.159|**1093K**|**259MB**|23m ~~28m~~|**5.887ms**|
> >
> > > ### Deep Blending dataset
> > |Method|PSNR ↑|SSIM ↑|LPIPS ↓|Prim ↓|Mem ↓|Training Time ↓|Rendering Time ↓|
> > |-|-|-|-|-|-|-|-|
> > |AbsGS|29.500|0.900|0.237|1961K|464MB|**27m**|7.353ms|
> > |AbsGS (60K iter.)|29.275|0.895|0.240|1962K|464MB|57m|7.549ms|
> > |AbsGS + DC4GS|**29.654**|**0.905**|**0.235**|**1499K**|**354MB**|40m ~~50m~~|**6.627ms**|

---

> > ### Comment · Reviewer_1vr6 · 2025-08-03
> >
> > Thanks for your answers. I think there might be an error. In original GS, a Gaussian with large gradient will be split when its scaling is larger than the threshold, and will be cloned when its scaling is smaller. I think Agorithm 1 in your paper conflicts with what you have written. However, I think this is an ignorable mistake. The modest improvements shown by most of your additional experiments still cannot address my concerns about the effectiveness of the proposed methods. Therefore, I decide to maintain my rating.

---

> > > ### Comment · Area_Chair_KMJK · 2025-08-05
> > >
> > > Some other reviewers lean towards positive, even though there is a marginal improvement with negligible rendering time overhead, due to their clear motivation and the inherent limitation of contribution from the Adaptive Density Control. After reading their reviews, do you still want to defend your position? If you leave your opinions on the other reviews, it'll help for a fair assessment.
> > >
> > > For the authors, would you like to provide feedback on this, advocating for your work before the author-reviewer discussion period ends?

---

> > > > ### Author Response · Authors · 2025-08-05
> > > >
> > > > Thank you very much for your thoughtful engagement and for giving us the opportunity to further clarify our position.
> > > >
> > > > We would be glad to continue the discussion. We believe this exchange is valuable, and we welcome the chance to advocate for the contribution and address any remaining concerns with more context.
> > > >
> > > > Thank you again for your time and consideration.

---

> > > ### Author Response · Authors · 2025-08-07
> > > **Sincerely expecting further discussions with Reviewer 1vr6**
> > >
> > > Dear Reviewer 1vr6,
> > >
> > > We sincerely thank you for the time and effort you have dedicated to reviewing our submission. Your thoughtful comments and observations have been instrumental in helping us strengthen our work.
> > >
> > > Following your earlier response, we have provided additional clarifications and experimental results to further address your concerns regarding the effectiveness of DC4GS. In particular, we included new results focusing on geometrically challenging regions and depth consistency, which further support the practical advantages of our approach.
> > >
> > > As we near the end of the discussion period, we would greatly appreciate it if you could let us know whether our responses have sufficiently addressed your concerns, or if there are any remaining issues we can further clarify.
> > >
> > > Thank you once again for your thoughtful review and valuable time.
> > >
> > > Best regards,
> > >
> > > The Authors of Submission 8673

---

> ### Author Response · Authors · 2025-08-05
>
> We sincerely thank the reviewer for the careful reading and thoughtful response to our rebuttal.
>
> We would like to take this opportunity to further clarify a few points and emphasize the strengths of DC4GS, especially in scenarios where our method meaningfully improves upon standard ADC, such as structurally challenging regions and depth accuracy.
>
> ## Correction on Split Condition
> You are absolutely right. The inequality direction in our rebuttal was incorrect.
> We apologize for the confusion and confirm that Algorithm 1 in the main paper is correct.
> Thank you again for helping us improve the paper’s clarity.
>
> ## Effectiveness in Challenging Regions
> While overall metric gains may appear modest, DC4GS shows clear advantages in **structurally challenging and perceptually sensitive regions** where standard ADC falls short.
> In the Truck scene, DC4GS accurately reconstructs features **behind transparent windows** (e.g., steering wheel) missed by AbsGS.
> In the Dr. Johnson scene, which contains **high-frequency textures** like carpet and wood scratches, DC4GS achieves notably better metrics.
> |Scene (Dataset)|Region Type|Method|PSNR ↑|SSIM ↑|LPIPS ↓|
> |-|-|-|-|-|-|
> |Truck (Tanks&Temples)|Transparent Window|AbsGS|20.875|0.621|0.333|
> |||AbsGS + DC4GS|**21.098**|**0.642**|**0.301**|
> |Dr Johnson (Deep Blending)|Strong Texture|AbsGS|25.741|0.717|0.334|
> |||AbsGS + DC4GS|**26.462**|**0.761**|**0.294**|
>
> These results show that DC4GS achieves improved fidelity in these specific regions under the tested conditions.
>
> ## Effectiveness in Geometric Consistency
>
> In addition to image-space metrics, we evaluated how DC4GS improves geometric consistency, particularly in terms of depth accuracy.
>
> As qualitative visualization is restricted in the rebuttal phase, we instead computed depth maps from rendered scenes and compared them against monocular depth estimations from the RGB images using Depth Anything v2 [6]. This provides a consistent pseudo-ground-truth reference.
> We evaluated the resulting depth maps using standard depth metrics: RMSE, SSIM, threshold accuracy (δ < 1.25, δ < 1.25², δ < 1.25³)
>
> | Method|RMSE ↓| SSIM ↑|δ < 1.25 ↑|δ < 1.25² ↑|δ < 1.25³ ↑|
> |-|-|-|-|-|-|
> |AbsGS|0.277|0.703|0.126|0.272|0.479|
> |AbsGS + DC4GS (ours)|**0.274**|**0.707**|**0.129**|**0.283**|**0.491**|
> |GT RGB (reference)|0.000|1.000|1.000|1.000|1.000|
>
> These results suggest that DC4GS yields reconstructions that are not only more accurate visually, but also structurally more faithful in terms of depth, as reported in the Depth Anything v2[6] paper itself, **even small differences in δ metrics reflect meaningful structural changes**, indicating non-negligible structural improvements.
> Although visual depth comparisons are not allowed during the rebuttal stage, we observed qualitatively sharper and more coherent depth maps, and we will include these in the final manuscript revision.
>
> ## Generalization to Diverse Baselines and Scenes
>
> As discussed in our initial rebuttal, DC4GS consistently improves performance in **static**, **dynamic**, and **architecture-diverse** scenarios:
>
> - Integrated into 3DGS, AbsGS, Pixel-GS, GES, Scaffold-GS, 4DGS, and LPM [7];
> - Robust on **dynamic scene**, with up to **+1.2 dB PSNR**;
> - Consistently reduces **memory usage** with comparable or superior quality.
>
> These results confirm DC4GS’s wide applicability, even under **temporally changing** or **non-rigid** motion.
>
> ## Longer Training Does Not Substitute for DC4GS
>
> As a reinforcement of the experiment already presented in our previous rebuttal, we would like to **re-emphasize** that **prolonged training does not resolve the limitations of conventional ADC splitting**.
>
> To test this, we trained the baseline AbsGS for **60K iterations** (2× the default), allowing ample time for sub-primitives to adjust.
> However, results clearly show:
>
> - **DC4GS still outperformed** the 60K-trained baseline
> - In some cases (e.g., Deep Blending), longer training even degraded the performance due to compounding effects of misplacement
>
> **Conclusion**:
> - DC4GS’s improvements stem from **correct and coherent sub-primitive placement at split-time**, not brute-force optimization.
> - Directional Consistency provides **a meaningful inductive bias** that standard ADC lacks — and cannot recover from via brute force.
>
>
> We deeply appreciate your thoughtful and rigorous review.
> We believe these additional clarifications further demonstrate that:
>
> - DC4GS is a generalizable scheme that explicitly accounts for structural complexity.
> - Its effectiveness **extends beyond simple metric gains** into challenging conditions.
>
> Your detailed feedback helped us greatly strengthen the paper.
> Thank you again.
>
> ## References
> [6] Yang, Lihe, et al. "Depth anything v2." Advances in Neural Information Processing Systems 37 (2024): 21875-21911
> [7] Yang, Haosen, et al. "Improving Gaussian Splatting with Localized Points Management." Proceedings of the Computer Vision and Pattern Recognition Conference. 2025.

---

### Official Review · Reviewer_7zgh · 2025-07-01

**Clarity:** 3
**Significance:** 3
**Originality:** 3
**Rating:** 5
**Confidence:** 4

**Summary:**

This work focuses on the limitations in 3D Gaussian Splatting (3DGS) scene reconstruction via Adaptive Density Control (ADC), specifically the redundant segmentation and random segmentation positions arising from ADC's reliance on positional gradient magnitude for primitive splitting. To address this issue, this paper proposes Directional Consistency-driven Adaptive Density Control (DC4GS), introducing Directional Consistency (DC) as a structural-aware metric to evaluate gradient angular coherence, and further designs a DC-weighted Split Criterion (DCC) for adaptive splitting decisions and a DC-guided Split (DCS) scheme for optimal sub-primitive placement.

**Questions:**

1) ​​Does the method work on different scenarios (e.g., strong textures, transparent objects)?​​
​​2) Is the current method applicable to dynamic scenes (e.g., object motion, camera movement)? Could DC during dynamic changes lead to wrong splitting?

**Ethical Concerns:**

["NO or VERY MINOR ethics concerns only"]

**Final Justification:**

The rebuttal resolves most of my concerns, so I maintain my original rating (Accept).

**Limitations:**

yes.

**Paper Formatting Concerns:**

No Paper Formatting Concerns.

**Quality:**

3

**Strengths And Weaknesses:**

Strengths：
1) Enhanced Reconstruction Fidelity​. In addtion to reducing the number of primitives, DC4GS can also significantly enhance reconstruction fidelity. It better preserves high-frequency details such as fine structures like window frames and railings, avoiding the common issues of over-smoothing or structural discontinuities in traditional methods.
2) Primitive Reduction. By introducing Directional Consistency (DC) to drive adaptive density control, DC4GS can effectively avoid redundant segmentation in structurally simple regions, which significantly decrease memory usage.
3) Extensive experiments conducted on three datasets (Mip-NeRF360, Tanks&Temples, Deep Blending), covering indoor/outdoor and high/low frequency scenes. Three 3DGS methods (original 3DGS, AbsGS, Pixel-GS) were integrated as baselines to ensure the universality of the conclusions.

Weaknesses:
1) DC calculation per primitive increases per-iteration cost.
2) There is no significant improvement in quantitative indicators.

---

> ### Author Rebuttal · Authors · 2025-07-31
>
> ## Generalizability to Diverse Scene Characteristics
>
> >**Questions 1**
> > 1\) Does the method work on different scenarios (e.g., strong textures, transparent objects)?​​​​
> > 2\) Is the current method applicable to dynamic scenes (e.g., object motion, camera movement)? Could DC during dynamic changes lead to wrong splitting?
>
> We thank the reviewer for this important question.
>
> To assess the robustness of DC4GS across diverse scenarios, we performed additional region-level analyses on challenging visual cases from our existing datasets (Mip-NeRF360, Tanks & Temples, Deep Blending), and further evaluated its behavior on dynamic scenes (D-NeRF [4], DyNeRF [5]) datasets using 4DGS [3].
>
> ### 1) Applicability to Challenging Scenarios (e.g., Strong Textures, Transparent Objects)
>
> To evaluate DC4GS under challenging scene conditions, we perform a region-specific analysis on two representative cases involving transparency and strong textures.
>
> For the Truck scene in Tanks & Temples (4th image), we focused on the transparent window area. AbsGS fails to reconstruct the internal features behind the glass (e.g., steering wheel), while DC4GS successfully preserves these structures.
>
> For the Dr. Johnson scene in Deep Blending (9th image), which includes strong (high-frequency) textures (carpet, wood scratches), DC4GS achieves superior metrics over AbsGS, confirming its strength in capturing fine details.
>
> |Scene (Dataset)|Region Type|Method|PSNR ↑|SSIM ↑|LPIPS ↓|
> |-|-|-|-|-|-|
> |Truck (Tanks&Temples)|Transparent Window|AbsGS|20.875|0.621|0.333|
> |||AbsGS + DC4GS|**21.098**|**0.642**|**0.301**|
> |Dr Johnson (Deep Blending)|Strong Texture (Carpet)|AbsGS|25.741|0.717|0.334|
> |||AbsGS + DC4GS|**26.462**|**0.761**|**0.294**|
>
> These results demonstrate that DC4GS performs robustly under high-frequency and transparent conditions.
>
> ### 2) Applicability to Dynamic Scenes
>
> To further evaluate the applicability to dynamic scene, we applied DC4GS to 4DGS [3] on D-NeRF [4] and DyNeRF [5]. Notably, it yields not only reduced memory usage but also consistent PSNR improvements — up to **1.2 dB** in DyNeRF. These improvements confirm that DC4GS operates effectively even under dynamic and temporally varying settings.
>
> > ### D-NeRF and DyNeRF dataset
> | Dataset | Method | PSNR ↑ | SSIM ↑ | LPIPS ↓ | Prim. ↓ | Mem. ↓ | Training Time ↓ | Rendering Time ↓ |
> |---|---|---|---|---|---|---|---|---|
> | D-NeRF [4] | 4DGS | 34.063 | **0.978** | **0.026** | 45K | 10MB | **9m** | 15.233ms |
> |        | 4DGS + DC4GS | **34.124** | **0.978** | **0.026** | **40K** | **9MB** | 11m | **14.531ms** |
> | DyNeRF [5] | 4DGS | 29.718 | 0.930 | 0.153 | 123K | 29MB | **50m** | **32.137ms** |
> |        | 4DGS + DC4GS | **30.983** | **0.937** | **0.148** | **118K** | **28MB** | 1h 13m | 32.275ms |
>
>
>
> ## Runtime Overhead of DC Calculation
> >**Weakness 1**
> > DC calculation per primitive increases per-iteration cost.
>
> We thank the reviewer for raising the concern about the additional operations introduced by the Directional Consistency (DC)-driven Adaptive Density Control (ADC). While it is true that DC computation introduces overhead during training, we emphasize that this overhead is confined to the optimization phase and does not affect the rendering stage. In fact, DC4GS's reduction in primitive count yields faster rendering timing. We will move timing details from the supplementary to the main paper. Additionally, we will include a detailed per-iteration breakdown of the training time for each component, including DC calculation, DC-based split cost computation, and sub-primitive generation.
> Below is a breakdown of the overhead (measured on the Mip-NeRF360 bicycle scene with 44,206 primitives):
>
> |Module|Time (ms)|
> |-|-|
> |Directional Consistency (DC) Evaluation|0.01|
> |DC-based Split Cost Computation|31.02|
> |Sub-primitive Generation|16.913|
> |**Total Additional Overhead**|**47.943**|
>
> Among these, the DC-based split cost computation accounts for the largest portion of overhead, primarily due to atomic operations required during parallel processing. We will clarify these findings in the revised manuscript to ensure transparency regarding runtime performance.
>
> ## Quantitative Improvements
> >**Weakness 2**
> > There is no significant improvement in quantitative indicators.
>
> We thank the reviewer for pointing out the modest quantitative gains. DC4GS is designed to achieve a strong balance between rendering quality and resource efficiency, rather than solely maximizing metric scores. Across all benchmarks, DC4GS consistently achieves a substantial **10–30% reduction in both the number of primitives and memory usage**, while **maintaining or even slightly improving** visual fidelity.
>
> Importantly, the effectiveness of DC4GS extends beyond the baselines originally reported in the paper. We have successfully integrated DC4GS into several diverse 3DGS methods, Scaffold-GS [1], GES [2], and 4DGS [3]. For Scaffold-GS + DC4GS, since Scaffold-GS does not include split operation, we apply only the Directional Consistency-weighted split Criterion (DCC). In most cases, DC4GS demonstrated benefits in performance and memory efficiency.
>
> Furthermore, these improvements are not confined to static scenes. On dynamic datasets such as D-NeRF and DyNeRF, where 4DGS serves as the baseline, DC4GS delivers even greater gains in rendering quality (up to **+1.2 dB PSNR**), continuing to reduce memory footprint.
>
> These findings collectively reinforce the generality and robustness of DC4GS, showing that its contribution goes beyond marginal metric gains and proves valuable in a **broad range of scenarios and 3DGS architectures**.
>
> > ### Mip-NeRF360 dataset
> |Method|PSNR ↑|SSIM ↑|LPIPS ↓|Prim ↓|Mem ↓|Training Time ↓|Rendering Time ↓|
> |-|-|-|-|-|-|-|-|
> |Scaffold-GS [1]|**27.578**|0.809|0.225|569K|171MB|**49m**|9.244ms|
> |Scaffold-GS + DC4GS|27.577|**0.81**|0.225|**425K**|**128MB**|50m|**8.537ms**|
> |GES [2]|27.04|0.796|0.248|1543K|365MB|**32m**|6.981ms|
> |GES + DC4GS|**27.09**|**0.797**|0.248|**1323K**|**313MB**|42m|**6.698ms**|
>
> > ### Tanks&Temples dataset
> |Method|PSNR ↑|SSIM ↑|LPIPS ↓|Prim ↓|Mem ↓|Training Time ↓|Rendering Time ↓|
> |-|-|-|-|-|-|-|-|
> |Scaffold-GS [1]|**24.018**|**0.851**|**0.174**|247K|74MB|**29m**|7.015ms|
> |Scaffold-GS + DC4GS|24.016|0.849|0.177|**208K**|**62MB**|31m|**6.933ms**|
> |GES [2]| **23.64** |**0.842**|**0.191**|930K|220MB|**18m**|5.593ms|
> |GES + DC4GS|23.515|0.841|0.194|**816K**|**193MB**|22m|**5.441ms**|
>
> > ### Deep Blending dataset
> |Method|PSNR ↑|SSIM ↑|LPIPS ↓|Prim ↓|Mem ↓|Training Time ↓|Rendering Time ↓|
> |-|-|-|-|-|-|-|-|
> |Scaffold-GS [1]|**30.252**|**0.904**|**0.255**|171K|51MB|**42m**|6.298ms|
> |Scaffold-GS + DC4GS|30.063|0.903|0.257|**135K**|**40MB**|44m|**5.748ms**|
> |GES [2]|29.562|0.903|0.249|1606K|380MB|**40m**|6.835ms|
> |GES + DC4GS|**29.632**|**0.904**|**0.248**|**1487K**|**352MB**|45m|**6.571ms**|
>
>
> ## References
>
> [1] Lu, Tao, et al. "Scaffold-GS: Structured 3d gaussians for view-adaptive rendering." Proceedings of the IEEE/CVF Conference on Computer Vision and Pattern Recognition. 2024.
> [2] Hamdi, Abdullah, et al. "GES: Generalized exponential splatting for efficient radiance field rendering." Proceedings of the IEEE/CVF Conference on Computer Vision and Pattern Recognition. 2024.
> [3] Wu, Guanjun, et al. "4D gaussian splatting for real-time dynamic scene rendering." Proceedings of the IEEE/CVF conference on computer vision and pattern recognition. 2024.
> [4] Pumarola, Albert, et al. "D-NeRF: Neural radiance fields for dynamic scenes." Proceedings of the IEEE/CVF conference on computer vision and pattern recognition. 2021.
> [5] Li, Tianye, et al. "Neural 3d video synthesis from multi-view video." Proceedings of the IEEE/CVF conference on computer vision and pattern recognition. 2022.

---

> > ### Comment · Reviewer_7zgh · 2025-08-05
> >
> > I find that the additional experimental results provided in the rebuttal are inconsistent with those reported in the original paper — for example, the results of 4DGS on the DyNeRF dataset and Scaffold-GS on the Mip-NeRF360 dataset. If these rebuttal results are deemed incorrect, it suggests that the proposed method does not offer significant improvements.

---

> > > ### Author Response · Authors · 2025-08-06
> > > **Clarifying the Discrepancy in Reported 4DGS Results on DyNeRF**
> > >
> > > Thank you for raising concerns about the experimental results of 4DGS on the DyNeRF dataset.
> > > Below, we clarify the causes of the observed variability and explain our updated evaluation.
> > >
> > > ## Clarifications and Corrections
> > >
> > > 1. **High Variability in PSNR on DyNeRF**
> > > The PSNR scores on DyNeRF **fluctuate significantly** between training runs, even when using identical environments. This observation is also supported by the *Per-Gaussian Embedding-Based Deformation for Deformable 3D Gaussian Splatting (E-D3DGS)* paper [6], which reports PSNR values for 4DGS that differ substantially from those in the 4DGS paper [3].
> > > (Note: The method referred to as **"4DGaussians"** in the E-D3DGS [6] corresponds to what we denote as **"4DGS"** [3] in our rebuttal.)
> > >
> > > This discrepancy further confirms our observation that 4DGS results on DyNeRF  can vary significantly between runs.
> > > To provide more reliable results, we conducted **three independent training runs** for both 4DGS and 4DGS+DC4GS, and report the per-run and averaged results. Since PSNR showed high variance, we kindly suggest referring to **SSIM and LPIPS**, which were empirically much more stable over multiple training runs.
> > >
> > > 2. **Inconsistent LPIPS Metric Basis**
> > > Unlike other 3DGS-related works that use VGG-based LPIPS, the 4DGS paper [3] appears to report AlexNet-based LPIPS. To ensure consistency, we also report LPIPS values based on AlexNet.
> > >
> > > 3. **Memory Reporting Clarification**
> > > In our initial rebuttal, we mistakenly reported only the memory usage of point cloud primitives. However, 4DGS includes a **Gaussian deformation field network**, whose parameters also consume memory.
> > > We have corrected this and now report the **total memory** as the sum of primitive memory and deformation network parameter memory (coarse and fine stages). Our revised memory figures are **very close to those reported in the E-D3DGS paper [6]**, whereas the memory usage reported in the original 4DGS paper [3] appears to be higher than both of these values. It seems that 4DGS may have included additional components in their memory measurement, but we were unable to identify exactly what was included.
> > >
> > > The table below summarizes results from:
> > > - The original 4DGS paper [3];
> > > - The E-D3DGS paper [6], where 4DGS is referred to as **"4DGaussians"**;
> > > - Our own three independent runs of 4DGS and 4DGS+DC4GS, including averaged results.
> > >
> > > > #### DyNeRF Results from [3], [6], and Multi-Run Experiments
> > > | Dataset | Method | PSNR ↑ | SSIM ↑ | D-SSIM (1-MS-SSIM) ↓ | LPIPS (Alex) ↓ | Prim. ↓ | Mem. ↓ | Training Time ↓ | Rendering Time ↓ |
> > > |---|---|---|---|---|---|---|---|---|---|
> > > | DyNeRF [5] | 4DGS (from [3]) | 31.15 | 0.939 | 0.016 | 0.049 | - | 90MB | 40m | 33.3ms |
> > > | | 4DGS (from [6], denoted as **4DGaussians** in [6]) | 30.71 | 0.935 | - | 0.056 | - | 59MB | 50m | 19.2ms |
> > > | | 4DGS (Run 1) | 29.718 | 0.930 | 0.019 | 0.063 | 123K | 59MB | 50m | 32.137ms |
> > > | | 4DGS (Run 2) | 31.137 | 0.933 | 0.015 | 0.058 | 124K | 60MB | 50m | 33.247ms |
> > > | | 4DGS (Run 3) | 31.354 | 0.936 | 0.015 | 0.057 | 124K | 60MB | 50m | 32.228ms |
> > > | | **4DGS (Avg.)** | 30.736 | 0.933 | 0.016 | 0.059 | 123K | 59MB | **50m** | 32.537ms |
> > > | | 4DGS + DC4GS (Run 1) | 30.983 | 0.937 | 0.015 | 0.055 | 118K | 56MB | 1h 13m | 32.275ms |
> > > | | 4DGS + DC4GS (Run 2) | 30.836 | 0.934 | 0.015 | 0.057 | 120K | 57MB | 1h 13m | 32.241ms |
> > > | | 4DGS + DC4GS (Run 3) | 31.458 | 0.938 | 0.014 | 0.057 | 119K | 56MB | 1h 13m | 32.262ms |
> > > | | **4DGS + DC4GS (Avg.)** | **31.092** | **0.936** | **0.014** | **0.056** | **119K** | **56MB** | 1h 13m | **32.259ms** |
> > >
> > > Overall, these results still show that **DC4GS consistently improves quality with fewer primitive count**.
> > >
> > > We sincerely thank you for your detailed and constructive feedback.
> > > We hope that our clarifications and additional results have addressed your concerns and helped clarify the experimental differences.
> > >
> > > ## References
> > > [3] Wu, Guanjun, et al. "4D gaussian splatting for real-time dynamic scene rendering." Proceedings of the IEEE/CVF conference on computer vision and pattern recognition. 2024.
> > >
> > > [5] Li, Tianye, et al. "Neural 3d video synthesis from multi-view video." Proceedings of the IEEE/CVF conference on computer vision and pattern recognition. 2022.
> > >
> > > [6] Bae, Jeongmin, et al. "Per-gaussian embedding-based deformation for deformable 3d gaussian splatting." European Conference on Computer Vision. Cham: Springer Nature Switzerland, 2024.

---

> > > ### Author Response · Authors · 2025-08-06
> > > **Clarifying the Evaluation Discrepancy in Scaffold-GS**
> > >
> > > Thank you for pointing out the inconsistency in the additional results presented in the rebuttal.
> > > Below, we summarize the cause of the discrepancy and the corrective steps we have taken:
> > >
> > > The resolution used to evaluate Scaffold-GS in our rebuttal was not aligned with its official implementation.
> > > 1. **3DGS Resolution**
> > >    - For a fair comparison, we referred to the **officially provided rendered results available on the 3DGS authors' project page**.
> > >    - For each scene, we resized images to match the resolution used in those official results.
> > >
> > > 2. **Scaffold-GS Resolution**
> > >    - Scaffold-GS does **not resize** input images in its official implementation.
> > >    - However, in our rebuttal experiments, we mistakenly evaluated Scaffold-GS using the **downscaled images aligned with the 3DGS**.
> > >    - On the **Tanks and Temples** and **Deep Blending**, where resolution scaling is not required, our Scaffold-GS results are consistent with those in the paper.
> > >
> > > 3. **arXiv vs. CVPR Version of Scaffold-GS**
> > >    - If the reviewer referred the arXiv version of the Scaffold-GS paper, please note that it **does not include** the `flower` and `treehill` scenes from the Mip-NeRF360 dataset.
> > >    - We kindly ask you to refer to the CVPR published version of the Scaffold-GS paper [1], which includes all scenes in the evaluation.
> > >
> > > We have re-evaluated Scaffold-GS and ours using its resolution settings.
> > >
> > > > ### Mip-NeRF360 dataset
> > > |Method|PSNR ↑|SSIM ↑|LPIPS ↓|Prim ↓|Mem ↓|Training Time ↓|Rendering Time ↓|
> > > |-|-|-|-|-|-|-|-|
> > > |Scaffold-GS [1] (from paper [1])|27.720|0.811|0.228|-|171MB|-|9.803ms|
> > > |Scaffold-GS [1]|27.651|**0.810**|0.226|546K|164MB|**1h 1m**|9.705ms|
> > > |Scaffold-GS + DC4GS|**27.653**|**0.810**|**0.225**|**479K**|**144MB**|**1h 1m**|**9.563ms**|
> > >
> > > After correcting the resolution mismatch, the updated Scaffold-GS results closely match those reported in the original paper [1]. While there is still a slight difference in PSNR, the overall results are now nearly identical.
> > >
> > > Even under this corrected setting, the results still show that applying DC4GS consistently reduces the number of primitives and memory usage, while maintaining comparable or better quality.
> > > As noted in our previous rebuttal, since Scaffold-GS does not include a split operation, we apply only the Directional Consistency-weighted Criterion (DCC). These improvements are particularly meaningful, as they are achieved without increase in training time.
> > >
> > > We appreciate your careful attention to this matter and thank you again for helping us improve the rigor and fairness of our evaluation.
> > >
> > > ## References
> > > [1] Lu, Tao, et al. "Scaffold-GS: Structured 3d gaussians for view-adaptive rendering." Proceedings of the IEEE/CVF Conference on Computer Vision and Pattern Recognition. 2024.

---

> > > > ### Comment · Reviewer_7zgh · 2025-08-07
> > > >
> > > > The rebuttal resolves most of my concerns, so I maintain my original rating (Accept).

---

> > > > > ### Author Response · Authors · 2025-08-07
> > > > >
> > > > > Thank you for your thoughtful consideration and for taking the time to review our rebuttal. We are grateful that our clarifications were helpful. Please let us know if there are any remaining concerns we can further address.

---

> ### Author Response · Authors · 2025-08-05
>
> Thank you very much for your careful observation, and we sincerely apologize for any confusion caused.
>
> Each baseline and baseline+DC4GS results were obtained under identical environments and training settings, following the instructions from each method’s GitHub repository.
> We will thoroughly revisit the setups for 4DGS on DyNeRF and Scaffold-GS on Mip-NeRF360 to check if we may have missed any important details.
>
> We’ll verify the implementations and report back with either corrected or confirmed results as soon as possible.
> Thank you again for your attention, and we sincerely apologize once more for the oversight.

---

### Official Review · Reviewer_Gqr6 · 2025-07-03

**Clarity:** 3
**Significance:** 3
**Originality:** 3
**Rating:** 4
**Confidence:** 4

**Summary:**

The adaptive density control (ADC) in 3D Gaussian Splatting (3DGS) struggles to accurately represent the distribution of individual primitives, often resulting in redundant splits and misaligned or overlapping sub-primitives due to randomly chosen split positions. To address these limitations, this paper introduces a Directional Consistency (DC)-driven ADC. The proposed DC approach better captures local structural complexity and determines optimal split positions, ensuring that sub-primitives are well-aligned with the underlying geometry.

**Questions:**

See the Weaknesses.

**Ethical Concerns:**

["NO or VERY MINOR ethics concerns only"]

**Final Justification:**

The rebuttal experiments resolved some concerns. My rating remains the same.

**Limitations:**

Yes

**Quality:**

3

**Strengths And Weaknesses:**

**Strengths**:

(1) This paper offers a clear and logical explanation of the limitations arising from 3DGS's adaptive density control.

(2) The proposed method is compatible with various existing approaches and demonstrates strong effectiveness in practice.

**Weaknesses**:

(1) The improvements in rendering quality are marginal—less than 0.5 dB in PSNR, 0.02 in SSIM, and 0.03 in LPIPS—raising concerns about the practical effectiveness of the proposed method.

(2) Since the method introduces additional operations during the ADC stage, it is important to clarify whether this incurs extra training costs. The authors should provide a detailed analysis of both effectiveness and computational overhead.

(3) More qualitative comparisons are needed to clearly demonstrate the advantages of the Directional Consistency (DC)-driven ADC, particularly in terms of geometry visualization such as depth maps.

(4)Missing reference:
[1] Improving Gaussian Splatting with Localized Points Management (CVPR2025)

---

> ### Author Rebuttal · Authors · 2025-07-31
>
> ## On Marginal Improvements in Rendering Quality
>
> >**Weaknesses 1**
> > The improvements in rendering quality are marginal—less than 0.5 dB in PSNR, 0.02 in SSIM, and 0.03 in LPIPS—raising concerns about the practical effectiveness of the proposed method.
>
> We thank the reviewer for the helpful comment. While the absolute gains in PSNR, SSIM, and LPIPS may seem numerically modest, the main contribution behind DC4GS lies in its ability to enhance efficiency by significantly reducing the number of primitives and memory usage without sacrificing rendering quality. In practice, DC4GS achieves consistent 10–30% reductions in both memory and primitive count, all while maintaining or slightly improving visual fidelity.
>
> To further demonstrate its robustness and generality, we additionally have integrated DC4GS into multiple recent 3DGS pipelines, including Scaffold-GS [1], GES [2], and 4DGS [3], each employing different reconstruction and rendering strategies. In most cases, DC4GS either preserved or improved performance while achieving noticeable savings in memory and primitives. Notably, because **Scaffold-GS** does not involve any explicit primitive splitting logic, we only apply our DC-weighted Split Cost (DCC) module in this case, and omit the DC-guided Split (DCS). Even under this partial integration, the method demonstrates tangible efficiency gains.
>
> Most significantly, when applied to 4DGS, a method specifically developed for dynamic scene reconstruction, our approach not only reduced memory but also produced a substantial **+1.2 dB gain in PSNR** on the DyNeRF [5] dataset. This result is particularly significant as it suggests that DC4GS is not confined to static scenes, but also effective in temporally varying or dynamic settings, demonstrating its broad applicability and practical utility.
>
> The following results illustrate the consistency of DC4GS across various datasets and methods:
>
> > ### Mip-NeRF360 dataset
> |Method|PSNR ↑|SSIM ↑|LPIPS ↓|Prim ↓|Mem ↓|Training Time ↓|Rendering Time ↓|
> |-|-|-|-|-|-|-|-|
> |Scaffold-GS [1]|**27.578**|0.809|0.225|569K|171MB|**49m**|9.244ms|
> |Scaffold-GS + DC4GS|27.577|**0.81**|0.225|**425K**|**128MB**|50m|**8.537ms**|
> |GES [2]|27.04|0.796|0.248|1543K|365MB|**32m**|6.981ms|
> |GES + DC4GS|**27.09**|**0.797**|0.248|**1323K**|**313MB**|42m|**6.698ms**|
>
> > ### Tanks&Temples dataset
> |Method|PSNR ↑|SSIM ↑|LPIPS ↓|Prim ↓|Mem ↓|Training Time ↓|Rendering Time ↓|
> |-|-|-|-|-|-|-|-|
> |Scaffold-GS [1]|**24.018**|**0.851**|**0.174**|247K|74MB|**29m**|7.015ms|
> |Scaffold-GS + DC4GS|24.016|0.849|0.177|**208K**|**62MB**|31m|**6.933ms**|
> |GES [2]| **23.64** |**0.842**|**0.191**|930K|220MB|**18m**|5.593ms|
> |GES + DC4GS|23.515|0.841|0.194|**816K**|**193MB**|22m|**5.441ms**|
>
> > ### Deep Blending dataset
> |Method|PSNR ↑|SSIM ↑|LPIPS ↓|Prim ↓|Mem ↓|Training Time ↓|Rendering Time ↓|
> |-|-|-|-|-|-|-|-|
> |Scaffold-GS [1]|**30.252**|**0.904**|**0.255**|171K|51MB|**42m**|6.298ms|
> |Scaffold-GS + DC4GS|30.063|0.903|0.257|**135K**|**40MB**|44m|**5.748ms**|
> |GES [2]|29.562|0.903|0.249|1606K|380MB|**40m**|6.835ms|
> |GES + DC4GS|**29.632**|**0.904**|**0.248**|**1487K**|**352MB**|45m|**6.571ms**|
>
> > ### D-NeRF and DyNeRF dataset
> |Dataset|Method|PSNR ↑|SSIM ↑|LPIPS ↓|Prim ↓|Mem ↓|Training Time ↓|Rendering Time ↓|
> |-|-|-|-|-|-|-|-|-|
> |D-NeRF [4]|4DGS|34.063|**0.978**|**0.026**|45K|10MB|9m|15.233ms|
> ||4DGS + DC4GS |**34.124**|**0.978**|**0.026**|**40K**|**9MB**|11m|**14.531ms**|
> |DyNeRF [5]|4DGS|29.718|0.930|0.153|123K|29MB|50m|**32.137ms**|
> ||4DGS + DC4GS|**30.983**|**0.937**|**0.148**|**118K**|**28MB**|1h 13m|32.275ms|
>
> These results confirm that DC4GS can be seamlessly integrated into various 3DGS architectures across both static and dynamic scenarios, achieving meaningful efficiency improvements with superior or comparable quality.
>
> ## Computational Overhead
>
> > **Weakness 2**
> > Since the method introduces additional operations during the ADC stage, it is important to clarify whether this incurs extra training costs. The authors should provide a detailed analysis of both effectiveness and computational overhead.
>
> We appreciate this important concern. While DC4GS introduces additional computation during training, particularly for Directional Consistency (DC) evaluation and DC-based split cost computation, these operations are limited to the training phase only, and do not affect rendering time. In practice, DC4GS reduces the number of primitives, which accelerates the rendering time.
> We will move the training-time analysis (previously in supplementary) to the main paper and  include a breakdown of DC-related overhead:
>
> |Module|Time (ms)|
> |-|-|
> |Directional Consistency (DC) Evaluation|0.01|
> |DC-based Split Cost Computation|31.02|
> |Sub-primitive Generation|16.913|
> |**Total Additional Overhead**|**47.943**|
>
> These results are based on the MipNeRF360 "bicycle" scene with 44,206 primitives. Most overhead arises from atomic operations in DC-based splitting. We will include these findings in the revised version to clarify computational trade-offs.
>
> ## On the Lack of Qualitative Geometry Comparisons
> >**Weaknesses 3**
> > More qualitative comparisons are needed to clearly demonstrate the advantages of the Directional Consistency (DC)-driven ADC, particularly in terms of geometry visualization such as depth maps.
>
> We thank the reviewer for highlighting the need for stronger qualitative evidence of geometric improvements.
> To address this concern, we computed depth maps from both AbsGS and our method using a modified splatting pipeline, and compared them to evaluate the structural fidelity of the reconstructions.
>
> To extract depth maps from Gaussian Splatting, we adapt the original color rendering equation by replacing the RGB color contribution with per-Gaussian depth values. Specifically, instead of computing the rendered color as:
>
> $$ C = \sum_i T_i \cdot \alpha_i \cdot c_i,\quad \text{where} \quad T_i = \prod_{j=1}^{i-1}(1 - \alpha_j) $$
> where $c_i$ is the RGB color and $T_i$ is the accumulated transmittance, we compute the expected depth as:
> $$ D = \sum_i T_i \cdot \alpha_i \cdot d_i $$
>
> where $d_i$ is the depth of the $i$-th Gaussian in view or camera space.
>
> To obtain GT depth maps for evaluation, we used the Depth Anything v2 model to perform monocular depth estimation on the GT RGB images. We then used the depth map predicted from the GT RGB as a reference and compared the others against it using standard evaluation metrics:
> **RMSE**, **SSIM** and **threshold accuracy** (δ < 1.25, 1.25², 1.25³).
>
> The results, presented below, show that our **DC4GS** yields more geometrically consistent renderings, producing depth more aligned with the reference than the **AbsGS** on the Mip-NeRF360 dataset. We also observed qualitatively better results through depth map visualizations. Although we are unable to include these visualizations due to rebuttal guidelines, we plan to provide them in a revised version.
>
> > ### Depth Consistency Evaluation (MipNeRF360 dataset)
>
> ||RMSE ↓|SSIM ↑|δ < 1.25 ↑|δ < 1.25² ↑|δ < 1.25³ ↑|
> |-|-|-|-|-|-|
> |AbsGS|0.277|0.703|0.126|0.272|0.479|
> |AbsGS + DC4GS (ours)|**0.274**|**0.707**|**0.129**|**0.283**|**0.491**|
> |GT RGB (reference)|0.000|1.000|1.000|1.000|1.000|
>
> ## Missing Reference
> > **Weaknesses 4**
> > Missing reference: Improving Gaussian Splatting with Localized Points Management (CVPR2025)
>
> Thank you for pointing out the omission of the CVPR 2025 paper *"Improving Gaussian Splatting with Localized Points Management (LPM)"* [6]. We will add this reference to the revised Related Work section.
>
> To further assess the generality of our approach, we also applied DC4GS to the Localized Points Management method. The results of this experiment are presented in the table below and demonstrate that our method continues to yield benefits in quality and efficiency when combined with recent 3DGS variants.
>
> > ### Mip-NeRF360 dataset
> |Method|PSNR ↑|SSIM ↑|LPIPS ↓|Prim. ↓|Mem. ↓|Training Time ↓|Rendering Time ↓|
> |-|-|-|-|-|-|-|-|
> |LPM [6]|**27.589**|**0.820**|**0.212**|3426K|810MB|**38m**|10.702ms|
> |LPM + DC4GS|27.556|0.819|0.218|**2712K**|**641MB**|54m|**9.462ms**|
>
> > ### Tanks&Temples dataset
> |Method|PSNR ↑|SSIM ↑|LPIPS ↓|Prim. ↓|Mem. ↓|Training Time ↓|Rendering Time ↓|
> |-|-|-|-|-|-|-|-|
> |LPM [6]|23.878|**0.847**|**0.183**|1824K|431MB|**18m**|7.143ms|
> |LPM + DC4GS|**23.892**|0.846|0.186|**1502K**|**355MB**|27m|**6.37ms**|
>
> > ### Deep Blending dataset
> |Method|PSNR ↑|SSIM ↑|LPIPS ↓|Prim. ↓|Mem. ↓|Training Time ↓|Rendering Time ↓|
> |-|-|-|-|-|-|-|-|
> |LPM [6]|29.483|0.901|0.245|2525K|597MB|**30m**|9.09ms|
> |LPM + DC4GS|**29.584**|**0.903**|**0.244**|**2350K**|**555MB**|49m|**8.512ms**|
>
> ## References
>
> [1] Lu, Tao, et al. "Scaffold-GS: Structured 3d gaussians for view-adaptive rendering." Proceedings of the IEEE/CVF Conference on Computer Vision and Pattern Recognition. 2024.
> [2] Hamdi, Abdullah, et al. "GES: Generalized exponential splatting for efficient radiance field rendering." Proceedings of the IEEE/CVF Conference on Computer Vision and Pattern Recognition. 2024.
> [3] Wu, Guanjun, et al. "4D gaussian splatting for real-time dynamic scene rendering." Proceedings of the IEEE/CVF conference on computer vision and pattern recognition. 2024.
> [4] Pumarola, Albert, et al. "D-NeRF: Neural radiance fields for dynamic scenes." Proceedings of the IEEE/CVF conference on computer vision and pattern recognition. 2021.
> [5] Li, Tianye, et al. "Neural 3d video synthesis from multi-view video." Proceedings of the IEEE/CVF conference on computer vision and pattern recognition. 2022.
> [6] Yang, Haosen, et al. "Improving Gaussian Splatting with Localized Points Management." Proceedings of the Computer Vision and Pattern Recognition Conference. 2025.

---

> > ### Comment · Reviewer_Gqr6 · 2025-08-07
> >
> > Thank you for the rebuttal—it was very helpful. I will maintain my borderline accept decision.

---

> > > ### Author Response · Authors · 2025-08-08
> > >
> > > Thank you for your response and for considering our rebuttal. We truly appreciate your constructive feedback and the time you have devoted to reviewing our paper. Please do not hesitate to reach out if you have any further concerns or require additional clarification.

---

> ### Author Response · Authors · 2025-08-07
> **Sincerely Looking Forward to Continuing the Discussion with Reviewer Gqr6**
>
> Dear Reviewer Gqr6,
>
> We sincerely thank you again for your detailed and constructive review. Your thoughtful feedback has been incredibly helpful in shaping our revisions and addressing critical aspects of the paper.
>
> Following your suggestions, we conducted additional evaluations to address the concerns regarding the effectiveness of DC4GS, including:
>
> - Extended integration across recent 3DGS architectures (e.g., LPM, GES, Scaffold-GS, 4DGS),
>
> - Analysis of computational overhead during the ADC stage,
>
> - Depth-based consistency metrics comparing our method to baseline,
>
> - Inclusion of the missing reference to "Improving Gaussian Splatting with Localized Points Management."
>
> We hope these additions have helped clarify the contributions and practical impact of our method. As we approach the conclusion of the discussion period, we would greatly appreciate it if you could kindly let us know whether our responses have addressed your concerns, or if there are any remaining issues we can clarify further.
>
> Thank you once again for your thoughtful review and time.
>
> Best regards,
>
> The Authors of Submission 8673

---

### Official Review · Reviewer_yCGb · 2025-07-03

**Clarity:** 3
**Significance:** 3
**Originality:** 3
**Rating:** 4
**Confidence:** 3

**Summary:**

This paper presents DC4GS, a directional consistency-driven adaptive density control method for 3D Gaussian Splatting (3DGS). By integrating directional consistency into both split decisions and sub-primitive placement, DC4GS effectively captures scene geometry while reducing redundant computation. Experimental results demonstrate that DC4GS reduces the number of primitives while simultaneously improving novel view synthesis quality across diverse scenes.

**Questions:**

Numerous storage-efficient Gaussian Splatting methods have been proposed in recent years (especially 2024 and 2025). Although some are briefly mentioned in the related work section, they are not included in the experimental comparisons. The current experiments only include two baselines from 2024 along with older methods. It would be more appropriate to compare against more recent approaches, such as Scaffold-GS, to provide a more comprehensive evaluation.

**Ethical Concerns:**

["NO or VERY MINOR ethics concerns only"]

**Final Justification:**

The core idea of the paper is clear and reasonable. The authors addressed my concerns during the rebuttal period, demonstrating a better trade-off between quality and storage requirements compared to some state-of-the-art methods. Therefore, I update my final rating to positive.

**Limitations:**

Yes

**Paper Formatting Concerns:**

No formatting concern.

**Quality:**

2

**Strengths And Weaknesses:**

Strengths:
The paper is well-written, and the core idea is clear and reasonable. The experimental results support the contributions claimed by the authors.

Weaknesses:
- Since the proposed method requires additional computation during training as a trade-off, it is recommended to move the discussion on training costs from the supplementary material to the main paper.
- Numerous storage-efficient Gaussian Splatting methods have been proposed in recent years (especially 2024 and 2025). Although some are briefly mentioned in the related work section, they are not included in the experimental comparisons. The current experiments only include two baselines from 2024 along with older methods. It would be more appropriate to compare against more recent approaches, such as Scaffold-GS, to provide a more comprehensive evaluation.

---

> ### Author Rebuttal · Authors · 2025-07-31
>
> ## Regarding Training Cost Analysis
>
> >**Weaknesses 1**
> > Since the proposed method requires additional computation during training as a trade-off, it is recommended to move the discussion on training costs from the supplementary material to the main paper.
>
> We thank the reviewer for pointing out the importance of discussing the training cost of our method in the main paper. In response, we will move the training time analysis from the supplementary to the main manuscript and expand it with more detailed breakdowns.
>
> Specifically, we provide timing results for each major component in our pipeline, corresponding to the stages outlined in Algorithm 1: (1) Directional Consistency (DC) evaluation, (2) DC-based split cost computation, and (3) sub-primitive generation. The table below summarizes the per-iteration overhead measured on the bicycle scene from the Mip-NeRF360 dataset with 44,206 primitives:
>
> > ### Training Time Analysis
>
> |Module|Time (ms)|
> |-|-|
> |Directional Consistency (DC) Evaluation|0.01|
> |DC-based Split Cost Computation|31.02|
> |Sub-primitive Generation|16.913|
> |**Total Additional Overhead**|**47.943**|
>
> ## On Comparison with Recent Methods
>
> > **Weaknesses 2 and Questions 1**
> > It would be more appropriate to compare against more recent approaches, such as Scaffold-GS, to provide a more comprehensive evaluation.
>
> We appreciate the reviewer’s suggestion and agree that additional storage-efficient Gaussian Splatting methods should be included to strengthen the evaluation.
>
> To address this, we have extended our experiments to include Scaffold-GS [1] and Generalized Exponential Splatting (GES) [2].
> We integrated DC4GS into these pipelines as follows:
>
> - For Scaffold-GS + DC4GS, since Scaffold-GS does not include split operation, we apply only the Directional Consistency-weighted split Criterion (DCC) to modulate anchor growth. Even without the Directional Consistency-guided Split (DCS) step, this modification results in notable reductions in memory and primitive count, with rendering quality preserved at a comparable level.
> - For GES + DC4GS, we adopt both the DCC and DCS schemes. Consistent with other setups, the integration led to reductions in primitive count and memory usage, achieving comparable and superior rendering fidelity.
>
> > ### Mip-NeRF360 dataset
> |Method|PSNR ↑|SSIM ↑|LPIPS ↓|Prim ↓|Mem ↓|Training Time ↓|Rendering Time ↓|
> |-|-|-|-|-|-|-|-|
> |Scaffold-GS [1]|**27.578**|0.809|0.225|569K|171MB|**49m**|9.244ms|
> |Scaffold-GS + DC4GS|27.577|**0.81**|0.225|**425K**|**128MB**|50m|**8.537ms**|
> |GES [2]|27.04|0.796|0.248|1543K|365MB|**32m**|6.981ms|
> |GES + DC4GS|**27.09**|**0.797**|0.248|**1323K**|**313MB**|42m|**6.698ms**|
>
> > ### Tanks&Temples dataset
> |Method|PSNR ↑|SSIM ↑|LPIPS ↓|Prim ↓|Mem ↓|Training Time ↓|Rendering Time ↓|
> |-|-|-|-|-|-|-|-|
> |Scaffold-GS [1]|**24.018**|**0.851**|**0.174**|247K|74MB|**29m**|7.015ms|
> |Scaffold-GS + DC4GS|24.016|0.849|0.177|**208K**|**62MB**|31m|**6.933ms**|
> |GES [2]| **23.64** |**0.842**|**0.191**|930K|220MB|**18m**|5.593ms|
> |GES + DC4GS|23.515|0.841|0.194|**816K**|**193MB**|22m|**5.441ms**|
>
> > ### Deep Blending dataset
> |Method|PSNR ↑|SSIM ↑|LPIPS ↓|Prim ↓|Mem ↓|Training Time ↓|Rendering Time ↓|
> |-|-|-|-|-|-|-|-|
> |Scaffold-GS [1]|**30.252**|**0.904**|**0.255**|171K|51MB|**42m**|6.298ms|
> |Scaffold-GS + DC4GS|30.063|0.903|0.257|**135K**|**40MB**|44m|**5.748ms**|
> |GES [2]|29.562|0.903|0.249|1606K|380MB|**40m**|6.835ms|
> |GES + DC4GS|**29.632**|**0.904**|**0.248**|**1487K**|**352MB**|45m|**6.571ms**|
>
> Furthermore, to validate the scenario generality of our method beyond static reconstructions, we integrated DC4GS into 4DGS [3] designed for dynamic scene reconstruction. Despite the additional temporal complexity, our method yielded improvements in both **PSNR (+1.2dB)** and **efficiency**, demonstrating its robustness and applicability for a diverse settings.
>
> > ### D-NeRF and DyNeRF dataset
> | Dataset | Method | PSNR ↑ | SSIM ↑ | LPIPS ↓ | Prim ↓ | Mem ↓ | Training Time ↓ | Rendering Time ↓ |
> |---|---|---|---|---|---|---|---|---|
> | D-NeRF [4] | 4DGS | 34.063 | **0.978** | **0.026** | 45K | 10MB | 9m | 15.233ms |
> |        | 4DGS + DC4GS | **34.124** | **0.978** | **0.026** | **40K** | **9MB** | 11m | **14.531ms** |
> | DyNeRF [5] | 4DGS | 29.718 | 0.930 | 0.153 | 123K | 29MB | 49m | **32.137ms** |
> |        | 4DGS + DC4GS | **30.983** | **0.937** | **0.148** | **118K** | **28MB** | 1h 23m | 32.275ms |
>
> These results highlight that DC4GS not only complements existing methods aimed at storage efficiency, but also generalizes effectively to broader scenarios including dynamic scenes, validating its applicability, and performance gains.
>
> ## References
>
> [1] Lu, Tao, et al. "Scaffold-GS: Structured 3d gaussians for view-adaptive rendering." Proceedings of the IEEE/CVF Conference on Computer Vision and Pattern Recognition. 2024.
> [2] Hamdi, Abdullah, et al. "GES: Generalized exponential splatting for efficient radiance field rendering." Proceedings of the IEEE/CVF Conference on Computer Vision and Pattern Recognition. 2024.
> [3] Wu, Guanjun, et al. "4D gaussian splatting for real-time dynamic scene rendering." Proceedings of the IEEE/CVF conference on computer vision and pattern recognition. 2024.
> [4] Pumarola, Albert, et al. "D-NeRF: Neural radiance fields for dynamic scenes." Proceedings of the IEEE/CVF conference on computer vision and pattern recognition. 2021.
> [5] Li, Tianye, et al. "Neural 3d video synthesis from multi-view video." Proceedings of the IEEE/CVF conference on computer vision and pattern recognition. 2022.

---

> > ### Comment · Area_Chair_KMJK · 2025-08-05
> >
> > Dear Reviewer yCGb, the authors and AC want your feedback. Could you please leave a comment reflecting the rebuttal?

---

> > ### Comment · Reviewer_yCGb · 2025-08-06
> >
> > Thank you for your detailed response. All of my concerns have been addressed. I will update my rating in consideration of this rebuttal.

---

> ### Author Response · Authors · 2025-08-06
>
> Thank you very much for your thoughtful follow-up and for taking the time to review our rebuttal. We truly appreciate your constructive feedback throughout the review process. If there are any further clarifications or additional information we can provide, please do not hesitate to let us know.

---

### Note · Authors · 2025-08-11

We sincerely thank the reviewers and ACs for their constructive and insightful feedback, which has greatly helped refine our work. Based on the reviews, we have clarified the strengths of DC4GS, expanded experiments, and provided additional evidence addressing major concerns.

## Robustness in Static and Dynamic Scenes
DC4GS integrates seamlessly with diverse 3DGS variants, including 3DGS, AbsGS, Pixel-GS, Scaffold-GS, GES, 4DGS, and LPM, consistently delivering improved or comparable quality with fewer primitives. DC4GS remains effective in both static and dynamic scenes, showing clear gains in dynamic cases such as those handled by 4DGS, achieving higher quality over multiple runs with reduced memory usage. These results demonstrate the adaptability and robustness of DC4GS under varied scenarios and architectures.

## Geometric Fidelity
We evaluated geometric accuracy by comparing rendered depth maps with monocular depth estimates from test images (Mip-NeRF360 dataset). DC4GS consistently outperformed baselines in all metrics. Referring to related depth estimation literature, gains of this magnitude are regarded as meaningful, reinforcing that our improvements extend to the geometric structure.

## Performance in Challenging Visual Conditions
DC4GS shows clear advantages in regions with strong textures or transparency, where standard ADC often fails. Fine details such as intricate surface patterns, subtle scratches, or features behind transparent materials are preserved, avoiding the blurring or loss typically observed in baselines.

## Not Achievable by Longer Training Alone
We investigated whether simply extending training could match our results by doubling the default iterations of the baseline (AbsGS). Despite the additional training, the baseline still underperformed, and in the Deep Blending dataset, quality even degraded. These results indicate that our improvements stem from accurate, structural-complexity–aware splitting at split time, rather than brute-force optimization.

## Conclusion
DC4GS delivers high-quality rendering with fewer primitives in various 3DGS variants and scene types, including dynamic cases. Its directional consistency-driven adaptive density control enhances visual and geometric fidelity, preserving fine details in challenging regions. Extended training alone cannot match these results. We believe these results clearly validate the significance and practicality of DC4GS.

Thank you again for your time and consideration.

---

### Decision · Program_Chairs · 2025-09-17

**Decision:**

Accept (poster)

**Comment:**

They use the directional consistency of gradients using *circular mean* rather than simply summing up, arguing it avoids redundant splitting and is better for defining optimal split position.

Reviewer yCGb suggested including more storage-efficient methods to compare with state-of-the-art 3DGS methods to see if this work poses a new frontier in the respective field. The authors chose Scaffold-GS (Lu et al., CVPR 2024) and GES (Hamdi et al., CVPR 2024), showing an ablation study of how much their work DC4GS improves performance. On comparison with recent methods, although it is not explicitly mentioned during the rebuttal period, the authors also missed Octree-GS (Ren et al., 2024) and FLoD (Seo et al., 2024). For instance, Scaffold-GS+OctreeGS achieved 28.05 of PSNR and 140MB of Memory, while Scaffold-GS+DC4GS achieved 27.58 and 128MB, respectively, underperforming PSNR while having a similar memory footprint in Mip-NeRF360.

As Reviewers Gpr6 and 1vr6 repeatedly argued for the marginal improvement. 1) Fidelity is not significantly better or even worse in some cases, 2) # of primitives and memory footprints are reduced to some degree, but not significantly compared to other alternatives for memory reduction. 3) Training time is usually prolonged even though the number of primitives is reduced, 4) rendering time is slightly faster, but we do not know the confidence interval since the authors do not assess rigorously.

However, after author-reviewer discussions, Reviewer yCGb raises their score, while Reviewer 7zgh and Gqr6 hold positive opinions. AC agrees on that Reviewer 7zgh’s comment “the directional-consistency signal is simple and model-agnostic.” That’s why the proposed method could be orthogonal to the other storage-efficient methods. AC values its careful and insightful discussion on directional consistency of gradients and its applications to adaptive density control, although it has an inevitable trade-off among fidelity, parsimony, and time complexity.

Therefore, AC recommends this work for poster acceptance, but leaves room for adjustment depending on the overall slot allocation and SAC’s discretion. Good luck.

A minor comment for polishing the manuscript: According to the official NeurIPS style guideline, “all headings should be lower case (except for first word and proper nouns)” in Line 55 (style guide pdf). Please see Section 5 "Implementation *D*etails".